# Progressive Voronoi Diagram Subdivision Enables Accurate Data-free Class-Incremental Learning

**Chunwei Ma**[1], **Zhanghexuan Ji**[1], **Ziyun Huang**[2], **Yan Shen**[1], **Mingchen Gao**[1], **Jinhui Xu**[1]

[1]Department of Computer Science and Engineering, University at Buffalo
[2]Computer Science and Software Engineering, Penn State Erie
[1]`{chunweim,zhanghex,yshen22,mgao8,jinhui}@buffalo.edu`
[2]`{zxh201}@psu.edu`

## Abstract

Data-free Class-incremental Learning (CIL) is a challenging problem because rehearsing data from previous phases is strictly prohibited, causing catastrophic forgetting of Deep Neural Networks (DNNs). In this paper, we present *iVoro*, a novel framework derived from computational geometry. We found Voronoi Diagram (VD), a classical model for space subdivision, is especially powerful for solving the CIL problem, because VD itself can be constructed favorably in an incremental manner – the newly added sites (classes) will only affect the proximate classes, making the non-contiguous classes hardly forgettable. Furthermore, we bridge DNN and VD using Power Diagram Reduction, and show that the VD structure can be progressively refined along the phases using a divide-and-conquer algorithm. Moreover, our VD construction is not restricted to the deep feature space, but is also applicable to multiple intermediate feature spaces, promoting VD to be multilayer VD that efficiently captures multi-grained features from DNN. Importantly, *iVoro* is also capable of handling uncertainty-aware test-time Voronoi cell assignment and has exhibited high correlations between geometric uncertainty and predictive accuracy (up to ∼0.9). Putting everything together, *iVoro* achieves up to 25.26%, 37.09%, and 33.21% improvements on CIFAR-100, TinyImageNet, and ImageNet-Subset, respectively, compared to the state-of-the-art non-exemplar CIL approaches. In conclusion, *iVoro* enables highly accurate, privacy-preserving, and geometrically interpretable CIL that is particularly useful when cross-phase data sharing is forbidden, e.g. in medical applications.

## 1 Introduction

In many real-world applications such as medical imaging-based diagnosis, the learning system is usually required to be expandable to new classes, for example, from common to rare inherited retinal diseases (IRDs) (Miere et al., 2020), or from coarse to fine chest radiographic findings (Syeda-Mahmood et al., 2020), and importantly, without losing the knowledge already learned. This motivates the concept of *incremental learning* (IL) (Hou et al.,

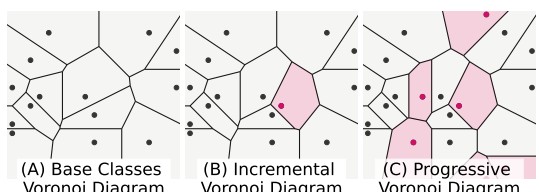

(A) Base Classes Voronoi Diagram  (B) Incremental Voronoi Diagram  (C) Progressive Voronoi Diagram

Figure 1: Schematic illustrations of Voronoi Diagram (VD) for base sites (A), and when a new site (B) or a clique of new sites (C) is added to the system.

2019; Wu et al., 2019; Zhu et al., 2021; Liu et al., 2021b), also known as *continual learning* (Parisi et al., 2019; Delange et al., 2021; Chaudhry et al., 2019), which has drawn growing interest in recent years. Although Deep Neural Networks (DNNs) have become the *de facto* method of choice due to their extraordinary ability to learn from complex data, they still suffer from severe catastrophic

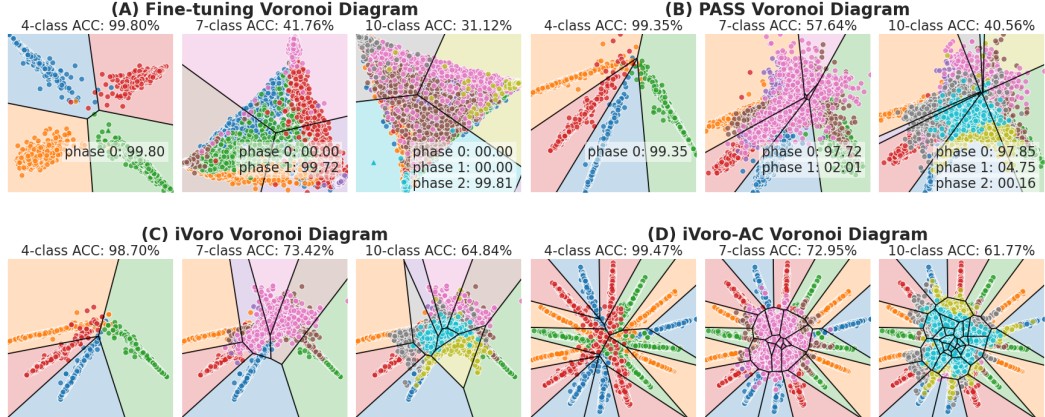

Figure 2: Visualization of of Voronoi Diagrams induced by (A) incremental fine-tuning, (B) PASS (Zhu et al., 2021), (C) iVoro, and (D) iVoro-AC on MNIST dataset in $\mathbb{R}^2$ (best viewed in color). The dataset was split to 4, 3, and 3 disjoint classes. (See Appendix B for details.)

forgetting (McCloskey & Cohen, 1989; Goodfellow et al., 2014; Kemker et al., 2018) when adapting to new tasks that contain *only* unseen training samples from novel classes.

To mitigate this issue, Rebuffi et al. (2017) proposed the paradigm of memory-based class-incremental learning (CIL) (Belouadah & Popescu, 2019; Zhao et al., 2020; Hou et al., 2019; Castro et al., 2018; Wu et al., 2019; Liu et al., 2021a; 2020a; 2021b) in which a small portion of samples (e.g., 20 exemplars per class) will be stored to use in the subsequent phases. However, the storing and sharing of data, e.g. medical images, may not be feasible due to privacy considerations. Another line of methods memorize (part of) network and increase the model capacity for new classes (Rusu et al., 2016; Li et al., 2019; Wang et al., 2017; Yoon et al., 2017), which may incur unbounded memory consumption for long task sequence. *Hence, in this paper, we focus on the challenging data-free CIL problem under the strictest memory and privacy constraints – no stored exemplars and fixed model capacity.*

Despite extensive research in recent years (see Appendix A for a literature review), three challenges still pose an obstacle to successful CIL. **(I)** During the course of isolated training on new data, the feature distributions of the old classes are usually dramatically changed (see Fig. 2 (A) for an illustration). Knowledge Distillation (KD) (Hinton et al., 2015) has become a routine in many CIL methods (Li & Hoiem, 2017; Schwarz et al., 2018; Castro et al., 2018; Hou et al., 2019; Dhar et al., 2019; Douillard et al., 2020; Zhu et al., 2021) to partially maintain the spatial distribution of old classes. The KD loss, however, is typically applied onto the whole network, and a strong KD loss may potentially degenerate the network's ability to adapt to novel classes. **(II)** Without the full access to old data, the decision boundaries cannot be learned precisely, making it harder to discriminate between old and new classes. Taking inspiration from metric-based Few-shot Learning (FSL) (Snell et al., 2017), PASS (Zhu et al., 2021) memorizes a set of prototypes (feature centroids) and generates features augmented by Gaussian noise for a joint training in new phases. However, feature centroids might be suboptimal to represent the whole class, which is not necessarily normally distributed (Fig. 2 (B)). **(III)** Since the old classes and the new classes are learned in a disjoint manner, their distributions are likely to be overlapped, which becomes even severer in our exemplar-free setting as the old data is totally absent. To circumvent this issue, *Task-incremental learning* (TIL) (Shin et al., 2017; Kirkpatrick et al., 2017; Zenke et al., 2017; Wu et al., 2018; Lopez-Paz & Ranzato, 2017; Buzzega et al., 2020; Cha et al., 2021; Pham et al., 2021; Fernando et al., 2017) assumes the phase within which a class was learned is known, which is generally unrealistic in practice. CIL is not grounded on this assumption.

In this paper, we tackle the CIL problem from a geometric point of view. Voronoi Diagram (VD) is a classical model for space subdivision and is the underlying geometric structure of the 1-nearest neighbor classifier (Lee, 1982). We find that VD bears a close analogy to incremental learning, because VD itself can be constructed favorably in an incremental manner – the newly added sites (classes) will roughly change only the cells of the neighboring classes, making the non-contiguous

classes untouched and thus hardly forgattable (see Figure 1). Based on this intuition, in this paper, we present a holistic geometric framework based on VD that significantly surmounts all the listed obstacles. The contributions can be summarized as follows:

**1.** We explore, for the first time, the idea of using prototypical networks (Snell et al., 2017) for CIL, which is equivalent to constructing a VD in the (fixed) feature space (denoted as iVoro).

**2.** We show that the within-phase boundaries of VD can be progressively refined using a divide-and-conquer algorithm (iVoro-D).

**3.** When it comes to test-time Voronoi cell assignment, we devise two protocols, augmentation consensus (iVoro-AC) and integration (iVoro-AI), for the postprocessing of Self-supervised Learning (SSL)-based label augmentation, with quantitative uncertainty awareness.

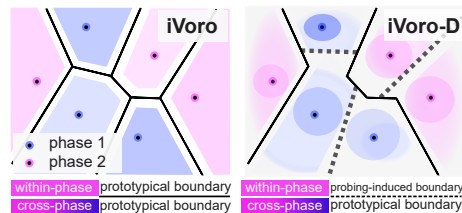

Figure 3: Schematic illustrations of iVoro (left) and iVoro-D (right). iVoro-D uses probing-induced boundaries within a phase.

**4.** Finally, we introduce multilayer features to build a multilayer VD, which consistently enhances the performance (iVoro-L).

**Main Ideas and Results**

*iVoro.* We begin with the simplest scenario in which the feature extractor is frozen after the first phase, and the prototypes ($\{c\}$) are used to construct VD.

*iVoro-D.* iVoro treats all prototypes equally regardless of at which phase they present, and determines Voronoi boundaries all by bisecting prototypes. However, without considering data distribution, the bisector of two prototypes is not optimal especially within a certain phase. We establish an explicit connection between DNN and VD using Voronoi Diagram Reduction (Ma et al., 2022a) and show that the within-phase disicion boundaries (induced by $\{\tilde{c}\}$) can be refined by DNN (i.e. linear probing) and be aggregated into the global VD by a divide-and-conquer (D&C) algorithm (iVoro-D).

*iVoro-AC/AI.* Geometrically, SSL-based label augmentation (Lee et al., 2020) will duplicate one Voronoi cell to be multiple (possibly disjoint) Voronoi cells (see Fig. 2 (D)), and this will cause ambiguity when assigning a query example to a cell, suggesting that uncertainty quantification cannot be neglected in test-time. Here we propose two protocols to resolve this ambiguity, namely, augmentation consensus (iVoro-AC) and augmentation integration (iVoro-AI). We also show that the entropy-based geometric variance (Ding & Xu, 2020) is a good indicator of the uncertainty of this assignment, with high Pearson correlation coefficients up to $\sim$0.9.

*iVoro-L.* Until now, only deep features from the last layer are used for VD construction. However, the intermediate feature could also be informative to aid the VD construction. Cluster-induced Voronoi Diagram (CIVD) (Chen et al., 2013; 2017; Huang & Xu, 2020; Huang et al., 2021), which allows for multiple centers per Voronoi cell, has recently achieved remarkable success in metric-based FSL by incorporating heterogeneous features to VD (Ma et al., 2022a). As a matter of fact, for a deep neural network, the feature induced by every layer can all be used to construct a VD. Finally, we also explore the idea to build a multilayer VD by using features elicited from multiple blocks of DNN.

*Broader impact.* The fully-fledged iVoro achieves up to 25.26%, 37.09%, and 33.21% improvements on CIFAR-100, TinyImageNet, and ImageNet-Subset, respectively, compared with the state-of-the-art non-exemplar CIL approaches. Based on the frozen model trained at the first phase, iVoro and all its variants incur no additional training burden, and at the same time preserve the privacy of data from previous phases. It is worth noting that, although iVoro focuses on exemplar-free CIL, it outperforms even all the exemplar-based CIL methods. We believe iVoro could be further boosted when a small number of exemplars are allowed, which we leave for future work.

## 2 METHODOLOGY

### 2.1 PRELIMINARIES: CLASS-INCREMENTAL LEARNING

In CIL, the data comes as a stream and a single model is trained on current data locally without revisiting previous data, but should ideally be able to discriminate between *all* classes it has seen so

far. Specifically, let $\mathcal{D} = \{\mathcal{D}_t\}_{t=1}^T$ be the data stream in which $\mathcal{D}_t = \{(\boldsymbol{x}_{t,i}, y_{t,i})\}_{i=1}^{N_t}$ is the dataset at time step $t$, with data $\boldsymbol{x}_{t,i} \in \mathbb{D}$ and label $y_{t,i} \in \mathcal{C}_t$. $\mathbb{D}$ is an arbitrary domain, e.g., natural image, and $\mathcal{C}_t$ is the set of classes at phase $t$. The dataset $\mathcal{D}_t$ contains $N_{t,k}, k \in \{1, ..., K_t\}$ samples for the $K_t$ classes (i.e. $N_t = \sum_{k=1}^{K_t} N_{t,k}$). Notice that $\mathcal{C}_i, \mathcal{C}_j$ for two arbitrary phases $i, j$ are disjoint, i.e. $\mathcal{C}_i \cap \mathcal{C}_j = \emptyset, \forall i, j : i \neq j$. The unified model consists of a feature extractor $\phi$ and a classification head $\theta$. The feature extractor is a deep neural network $\boldsymbol{z} = \phi(\boldsymbol{x}), \boldsymbol{z} \in \mathbb{R}^n$ that maps from image domain $\mathbb{D}$ to feature domain $\mathbb{R}^n$, and is (traditionally) trained continuously at each phase $t$. In this section, $T, t$, and $\tau$ denote total phase, current phase, and historical phase, respectively (i.e. $t \in \{1, ..., T\}$, $\tau \in \{1, ..., t\}$).

## 2.2 Constructing Voronoi Diagrams: A Feature Extractor is All You Need

In many CIL methods, the feature extractor $\phi$ and classification head $\theta$ are jointly and continuously optimized during every phase $t$ guided by carefully designed losses (Zhu et al., 2021). As a starting point, in this section, we freeze the feature extractor $\phi$ after the first phase and use a *Voronoi Diagram* (i.e., a 1-nearest-neighbor classifiers) to be $\theta$ as an extremely simple baseline method (denoted as iVoro), upon which we will then gradually add component introduced in Sec. 1. First, we introduce *Power Diagram* (PD), a generalized version of VD:

**Definition 2.1** (Power Diagram (Aurenhammer, 1987) and Voronoi Diagram). Let $\Omega = \{\omega_1, ..., \omega_K\}$ be a partition of the space $\mathbb{R}^n$, and $\mathcal{C} = \{\boldsymbol{c}_1, ..., \boldsymbol{c}_K\}$ be a set of centers (also called *sites*) such that $\cup_{r=1}^K \omega_r = \mathbb{R}^n, \cap_{r=1}^K \omega_r = \emptyset$. In addition, each center is associated with a weight $\nu_r \in \{\nu_1, ..., \nu_K\} \subseteq \mathbb{R}^+$. Then, the set of pairs $\{(\omega_1, \boldsymbol{c}_1, \nu_1), ..., (\omega_K, \boldsymbol{c}_L, \nu_K)\}$ is a Power Diagram (PD), where each cell is obtained via $\omega_r = \{\boldsymbol{z} \in \mathbb{R}^n : r(\boldsymbol{z}) = r\}, r \in \{1, .., K\}$, with $r(\boldsymbol{z}) = \arg\min_{k \in \{1, ..., K\}} d(\boldsymbol{z}, \boldsymbol{c}_k)^2 - \nu_k$. If the weights are equal for all $k$, i.e. $\nu_k = \nu_{k'}, \forall k, k' \in \{1, ..., K\}$, then a PD collapses to a Voronoi Diagram (VD).

**Prototypes.** As a baseline model, the class centers for iVoro are simply chosen to be the prototypes (feature mean of one class): $\boldsymbol{c}_{\tau,k} = \frac{1}{N_{\tau,k}} \sum_{i \in \{1, ..., N_{\tau,k}\}, y=k} \phi(\boldsymbol{x}_{\tau,i}), \nu_{\tau,k} = 0, \tau \in \{1, ..., t\}, k \in \{1, ..., K_\tau\}$. We name those centers *prototypical centers*. Note that this set of centers $\{\boldsymbol{c}_{\tau,k}\}$ carries prototypes for all classes, old and new, up to time $t$. In test-time, a query sample $\boldsymbol{x}$ is assigned to the nearest class $\hat{y} = \mathcal{C}_{\tau',k'}$ s.t. $d(\boldsymbol{z}, \boldsymbol{c}_{\tau',k'}) = \min_{\tau,k} d(\boldsymbol{z}, \boldsymbol{c}_{\tau,k})$ in which $d(\boldsymbol{z}, \boldsymbol{c}_{\tau,k}) = ||\boldsymbol{z} - \boldsymbol{c}_{\tau,k}||_2^2$.

**Parameterized Feature Transformation.** Although PASS (Zhu et al., 2021) uses Gaussian noise to augment the data, the actual features are not necessarily normally distributed. To encourage the normality of feature distribution here we adopt compositional feature transformation commonly used in FSL (Ma et al., 2022a): (1) $L_2$ *normalization* projects the feature onto the unit sphere: $f(\boldsymbol{z}) = \frac{\boldsymbol{z}}{||\boldsymbol{z}||_2}$; (2) *linear transformation* performs the scaling and shifting: $g_{w,\eta}(\boldsymbol{z}) = w\boldsymbol{z} + \eta$; and (3) *Tukey's ladder of powers transformation* further improves the Gaussianity: $h_\lambda(\boldsymbol{z}) = \begin{cases} \boldsymbol{z}^\lambda & \text{if } \lambda \neq 0 \\ \log(\boldsymbol{z}) & \text{if } \lambda = 0 \end{cases}$.

Finally, the feature transformation is the composition of three: $(h_\lambda \circ g_{w,\eta} \circ f)(\boldsymbol{z})$, parameterized by $w, \eta, \lambda$. If all features (for both training and testing set) go through this normalization function, then iVoro becomes iVoro-N.

## 2.3 Divide and Conquer: Progressive Voronoi Diagrams for CIL

As mentioned earlier, iVoro (and iVoro-N) treats all classes equally and separates them all by bisectors, regardless of at which phase they appear. However, for two classes $\mathcal{C}_{\tau,k_1}, \mathcal{C}_{\tau,k_2}$ appear in the same phase $\tau$, we can in fact draw better boundary by training a linear probing model parametrized by $\boldsymbol{W}, \boldsymbol{b}$ in the fixed feature space. After the training, the locating of new Voronoi center requires an explicit relationship between the probing model and VD. More formally, at phase $t$ a linear classifier with cross-entropy loss is optimized on the *local* data $\mathcal{D}_t$:

$$\mathcal{L}(\boldsymbol{W}_t, \boldsymbol{b}_t) = \sum_{(\boldsymbol{x},y) \in \mathcal{D}_t} -\log p(y|\phi(\boldsymbol{x}); \boldsymbol{W}_t, \boldsymbol{b}_t) = \sum_{(\boldsymbol{x},y) \in \mathcal{D}_t} -\log \frac{\exp(\boldsymbol{W}_{t,y}^T \phi(\boldsymbol{x}) + b_{t,y})}{\sum_k \exp(\boldsymbol{W}_{t,k}^T \phi(\boldsymbol{x}) + b_{t,k})} \quad (1)$$

in which $\boldsymbol{W}_{t,k}, b_{t,k}$ are the linear weight and bias for class $\mathcal{C}_{t,k}$. As a parameterized model, this linear probing can ideally improve the discrimination within $\mathcal{C}_t$. However, it is still non-trivial to merge all $\{\boldsymbol{W}_{\tau,k}, b_{\tau,k}\}_{\tau=1}^t$, since the task identity is not assumed to be known like in TIL. To solve this, we

get geometric insight from (Ma et al., 2022a) which directly connects linear probing model and VD by the theorem shown as follows:

**Theorem 2.1** (Voronoi Diagram Reduction (Ma et al., 2022a)). *The linear classifier parameterized by $\boldsymbol{W}, \boldsymbol{b}$ partitions the input space $\mathbb{R}^n$ to a Voronoi Diagram with centers $\{\tilde{\boldsymbol{c}}_1, ..., \tilde{\boldsymbol{c}}_K\}$ given by $\tilde{\boldsymbol{c}}_k = \frac{1}{2} \boldsymbol{W}_k$ if $b_k = -\frac{1}{4} ||\boldsymbol{W}_k||_2^2, k = 1, ..., K$.*

For completeness, we also include the proof in Appendix E. During linear probing, if Thm. 2.1 is satisfied, then it is guaranteed that the resulting centers (referred to as *probing-induced centers*) $\{\tilde{\boldsymbol{c}}_{t,k}\}_{k=1}^{K_t}$ will also induce a VD (locally in phase $t$). Now given that we have two sets of centers $\{\boldsymbol{c}_{\tau,k}\}$ and $\{\tilde{\boldsymbol{c}}_{\tau,k}\}$, with the latter being better locally but are not transferable across phases, we devise a divide-and-conquer (D&C) algorithm that progressively construct the decision boundaries from the two sets of centers, boosting iVoro to iVoro-D.

**Divide.** Fortunately, the total classes $\{\mathcal{C}_\tau\}_{\tau=1}^t$ have been split into already disjoint $t$ cliques.

**Conquer.** Within each clique (i.e. phase) $\tau$, the boundary for any two classes $\mathcal{C}_{\tau,k_1}, \mathcal{C}_{\tau,k_2}$ is the bisector separating the *probing-induced centers* $\tilde{\boldsymbol{c}}_{\tau,k_1}, \tilde{\boldsymbol{c}}_{\tau,k_2}$, denoted as $\Gamma_{\tau,k_1,\tau,k_2} = \{\boldsymbol{z}' \in \mathbb{R}^n | \boldsymbol{v}^T \boldsymbol{z}' - q = 0\}$ where $\boldsymbol{v} = \frac{\tilde{\boldsymbol{c}}_{\tau,k_1} - \tilde{\boldsymbol{c}}_{\tau,k_2}}{||\tilde{\boldsymbol{c}}_{\tau,k_1} - \tilde{\boldsymbol{c}}_{\tau,k_2}||_2}$ and $q = \frac{||\tilde{\boldsymbol{c}}_{\tau,k_1}||_2^2 - ||\tilde{\boldsymbol{c}}_{\tau,k_2}||_2^2}{2||\tilde{\boldsymbol{c}}_{\tau,k_1} - \tilde{\boldsymbol{c}}_{\tau,k_2}||_2}$. When merging cliques $\tau_1, \tau_2$, we instead resort to the *prototypical centers* for space partition: for any $\boldsymbol{c}_{\tau_1,k}$ in clique $\tau_1$ and any $\boldsymbol{c}_{\tau_2,k'}$ in clique $\tau_2$, their bisector is $\Gamma_{\tau_1,k,\tau_2,k'} = \{\boldsymbol{z}' \in \mathbb{R}^n | \boldsymbol{v}^T \boldsymbol{z}' - q = 0\}$ where $\boldsymbol{v} = \frac{\boldsymbol{c}_{\tau_1,k} - \boldsymbol{c}_{\tau_2,k'}}{||\boldsymbol{c}_{\tau_1,k} - \boldsymbol{c}_{\tau_2,k'}||_2}$ and $q = \frac{||\boldsymbol{c}_{\tau_1,k}||_2^2 - ||\boldsymbol{c}_{\tau_2,k'}||_2^2}{2||\boldsymbol{c}_{\tau_1,k} - \boldsymbol{c}_{\tau_2,k'}||_2}$. In this way, the overall space partition would benefit from both locally probing-induced VD and globally prototype-based VD. See Fig. G.3 for an illustrative comparison of iVoro and iVoro-D.

**Querying the VD.** In test-time, one can find the assigned Voronoi cell for query example $\boldsymbol{x}$ by eliminating one class in each round according to $\text{sign}(\boldsymbol{v}^T \boldsymbol{z}' - q)$, starting from a randomly selected boundary, so the time complexity is $\mathcal{O}(\sum_{\tau=1}^t K_\tau)$.

## 2.4 Augmentation Integration: Uncertainty-aware Test-time Voronoi Cell Assignment

**Self-supervised Label Augmentation.** To enhance the discriminative power of CIL method, SSL-based label augmentation (Lee et al., 2020) has been used to expand the original $K_t$ classes to $4K_t$ by rotating the original image $\boldsymbol{x}$. Specifically, for image $\boldsymbol{x}$, the rotated image $\boldsymbol{x}^{(\alpha)} = rotate(\boldsymbol{x}, \frac{\pi}{2}\alpha), \alpha \in \{0, 1, 2, 3\}$ will be assigned to one of the expanded classes $\hat{y} = k^{(\alpha)}, k \in \{1, ..., K\}, K \in \sum_{\tau=1}^t K_\tau$. In training time, the model is trained on the expanded dataset; however, in testing time, each of the duplicated images $\{\boldsymbol{x}^{(\alpha)}\}_{\alpha \in \{0,1,2,3\}}$ could possibly be assigned to each of the expanded classes $\{k^{(\alpha)}\}_{\alpha \in \{0,1,2,3\}}$, so this ambiguity has to be resolved, which has not been considered in previous CIL methods.

**Augmentation Consensus.** Let $\boldsymbol{d}^{(\alpha,\alpha')} \in \mathbb{R}^K$ be a vector, each component of which denotes the distance from $\phi(\boldsymbol{x})$ to a class that $\phi$ has learned, i.e. $d_k^{(\alpha,\alpha')} = d(\phi(\boldsymbol{x}^{(\alpha)}), \boldsymbol{c}_{k^{(\alpha')}}) = ||\phi(\boldsymbol{x}^{(\alpha)}) - \boldsymbol{c}_{k^{(\alpha')}}||_2^2, k \in \{1, ..., K\}, \alpha, \alpha' \in \{0, 1, 2, 3\}$. Then we want to find a consensus $\hat{k}$, with the maximum occurrence among the $4 \times 4$ predictions $\{\arg\min_k d_k^{(\alpha,\alpha')}\}_{\alpha,\alpha' \in \{0,1,2,3\}}$. Using augmentation consensus in test-time, iVoro is then retrofitted to iVoro-AC.

**Augmentation Integration.** Using the consensus from the augmented samples should be more robust than the individual prediction $\arg\min_k d_k^{(0,0)}$ itself, but it has not considered the accumulated distance, so alternatively, we propose to integral over all predictions from augmented samples:

$$\hat{k} = \arg\min_k \sum_\alpha \sum_{\alpha'} d_k^{(\alpha,\alpha')}.$$

If augmentation integration is applied, then iVoro becomes iVoro-AI.

**Uncertainty Quantification.** Since in iVoro-AC and iVoro-AI, the augmented samples collaboratively contribute to the final prediction, the quantitative uncertainty becomes non-negligible, this is because for some rotation-invariant classes, e.g. balls, the rotation operation makes less sense. Hence, when assigning a query sample $\boldsymbol{x}$ to the augmented $4\times$ Voronoi cells, an uncertainty quantification method is needed.

Truth Discovery Ensemble (TDE) (Ma et al., 2021) is the state-of-the-art uncertainty calibration method for DNNs, which finds the consensus among ensemble members by the minimization of entropy-based geometric variance (HV). Here, we only borrow HV as an indicator for the uncertainty of the $4 \times 4$ predictions, and refer the readers to Ma et al. (2021) for more details about TDE. Given the mean vector of the augmented predictions $\boldsymbol{d}^* = \frac{1}{16}\sum_{\alpha,\alpha'}\boldsymbol{d}^{(\alpha,\alpha')} \in \mathbb{R}^K$, let $V$ denote the total squared distance to $\boldsymbol{d}^*$ (i.e., $V = \sum_\alpha \sum_{\alpha'} ||\boldsymbol{d}^* - \boldsymbol{d}^{(\alpha,\alpha')}||^2$) and $q^{(\alpha,\alpha')}$ denotes the contribution of each $\boldsymbol{d}^{(\alpha,\alpha')}$ to $V$ (i.e., $q^{(\alpha,\alpha')} = ||\boldsymbol{d}^* - \boldsymbol{d}^{(\alpha,\alpha')}||^2/V$). Then the entropy induced by $\{q^{(\alpha,\alpha')}\}$ is:

$$H = -\sum_\alpha\sum_{\alpha'}q^{(\alpha,\alpha')} \log q^{(\alpha,\alpha')} = \frac{1}{V}\sum_\alpha\sum_{\alpha'}||\boldsymbol{d}^* - \boldsymbol{d}^{(\alpha,\alpha')}||^2 \log(V/||\boldsymbol{d}^* - \boldsymbol{d}^{(\alpha,\alpha')}||^2).$$

Based on these, we can define the HV as follows:

**Definition 2.2** (Entropy-based Geometric Variance (Ding & Xu, 2020)). Given the point set $\{\boldsymbol{d}^{(\alpha,\alpha')}\} \subseteq \mathbb{R}^K$ and a point $\boldsymbol{d}^*$, the entropy based geometric variance (HV) is $H \times V$ where $H$ and $V$ are defined as shown above.

For every query example $\boldsymbol{x}$, we calculate HV$(\boldsymbol{x})$ based on its $\{\boldsymbol{d}^{(\alpha,\alpha')}\}_{\alpha,\alpha' \in \{0,1,2,3\}}$. Later we will show how HV could favorably indicate the uncertainty of the augmented prediction, and tell us when augmentation integration is useful.

### 2.5 MULTILAYER VORONOI DIAGRAMS

Until now, our VD construction is restricted to the deep feature space, i.e., $\boldsymbol{x} \mapsto \phi(\boldsymbol{x}) \in \mathbb{R}^n$. However, the intermediate layers also contain information that supplementary to the final layer and can be useful to our VD construction. And this requires the integration of multiple VDs. Recently, Cluster-induced Voronoi Diagram (CIVD) (Chen et al., 2017; Huang et al., 2021) and Cluster-to-cluster Voronoi Diagram (CCVD) (Ma et al., 2022a), two advanced VD structures, have shown remarkable ability to integrate multiple sets of centers for VD construction and achieve state-of-the-art performance in metric-based FSL. In this paper, we utilize the concept of CCVD for the integration of multiple VDs induced by multiple layers. We refer the readers to Ma et al. (2022a) for more details about CIVD/CCVD.

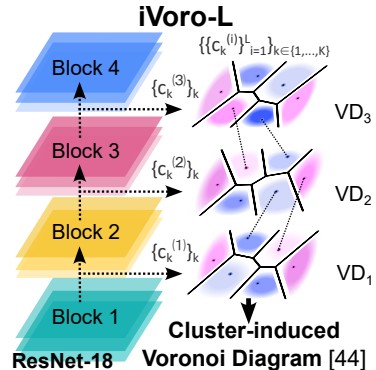

Figure 4: Schematic illustration of iVoro-L.

**Definition 2.3** (Cluster-to-cluster Voronoi Diagram). Let $\Omega = \{\omega_1, ..., \omega_K\}$ be a partition of the space $\mathbb{R}^n$, and $\mathcal{C} = \{\mathcal{C}_1, ..., \mathcal{C}_K\}$ be a set of *totally ordered sets* with the same cardinality $L$ (i.e. $|\mathcal{C}_1| = |\mathcal{C}_2| = ... = |\mathcal{C}_K| = L$). The set of pairs $\{(\omega_1, \mathcal{C}_1), ..., (\omega_K, \mathcal{C}_K)\}$ is a Cluster-to-cluster Voronoi Diagram (CCVD) with respect to an *influence function* $F(\mathcal{C}_k, \mathcal{C}(\boldsymbol{z}))$, and each cell is obtained via $\omega_r = \{\boldsymbol{z} \in \mathbb{R}^n : r(\boldsymbol{z}) = r\}, r \in \{1, .., K\}$, with $r(\boldsymbol{z}) = \arg\max_{k \in \{1,...,K\}} F(\mathcal{C}_k, \mathcal{C}(\boldsymbol{z}))$ where $\mathcal{C}(\boldsymbol{z})$ is the cluster (also a totally ordered set with cardinality $L$) that query point $\boldsymbol{z}$ belongs to, meaning that, all points in this cluster (query cluster) will be assigned to the same cell. The Influence Function is defined upon two totally ordered sets $\mathcal{C}_k = \{\boldsymbol{c}_k^{(i)}\}_{i=1}^L$ and $\mathcal{C}(\boldsymbol{z}) = \{\boldsymbol{z}^{(i)}\}_{i=1}^L$: $F(\mathcal{C}_k, \mathcal{C}(\boldsymbol{z})) = -\operatorname{sign}(\gamma) \sum_{i=0}^L d(\boldsymbol{c}_k^{(i)}, \boldsymbol{z}^{(i)})^\gamma$.

As CCVD is a flexible framework and can be applied to iVoro-D/AC/AI, here, as an example, we show how CCVD can be use to boost iVoro. In iVoro, the VD is induced by $\{\boldsymbol{c}_{\tau,k}\}_{\tau \in \{1,...,t\}, k \in \{1,...,K_\tau\}}$ that are feature means from the last layer $\phi$. Now, we arbitrarily extract $L$ layers $\{\phi^{(l)}\}_{l=1}^L$ and generate the $K$ totally ordered clusters $\{\{\boldsymbol{c}_{\tau,k}^{(l)}\}_{l=1}^L\}_{\tau \in \{1,...,t\}, k \in \{1,...,K_\tau\}}$ to construct CCVD and generate the query cluster $\{\phi^{(l)}(\boldsymbol{x})\}_{l=1}^L$ for the query example $\boldsymbol{x}$ for Voronoi cell assignment. See Appendix C for a summary of the notations and acronyms.

## 3 EXPERIMENTS

In our geometric framework, starting from iVoro, the simplest prototype-induced VD model, we gradually add four components: (**I**) parameterized normalization (iVoro-N), (**II**) divide-and-conquer

Table 1: Comparison between the fully-fledged iVoro with state-of-the-art non-exemplar (marked by ✘) and exemplar-based (marked by ✔) CIL methods in terms of the accuracy (in %) in the last phase and the average accuracy (Avg., in %) across all phases. *imp.↑* indicates the relative improvement upon the next best *non-exemplar* CIL method. ‡The best version of iVoro is shown here. See Tab. 2 for different versions of iVoro. Note that RMM uses a 100-class subset of ImageNet that is different from others (shown in blue).

| Methods | CIFAR-100 | | | | | | TinyImageNet | | | | | | ImageNet-Subset | |
|---|---|---|---|---|---|---|---|---|---|---|---|---|---|---|
| | 5 phases | | 10 phases | | 20 phases | | 5 phases | | 10 phases | | 20 phases | | 10 phases | |
| | Avg. | Last | Avg. | Last | Avg. | Last | Avg. | Last | Avg. | Last | Avg. | Last | Avg. | Last |
| ✔ iCaRL$_{CNN}$ (Rebuffi et al., 2017) | 51.25 | 40.50 | 48.52 | 39.13 | 44.85 | 34.38 | 34.90 | 23.20 | 31.12 | 20.82 | 28.03 | 20.20 | 50.61 | 38.40 |
| ✔ iCaRL$_{NCM}$ (Rebuffi et al., 2017) | 58.13 | 48.00 | 53.91 | 45.38 | 50.79 | 40.88 | 46.08 | 34.43 | 43.42 | 33.33 | 38.08 | 27.65 | 60.89 | 50.06 |
| ✔ EEIL (Castro et al., 2018) | 60.15 | 50.13 | 55.91 | 47.63 | 52.79 | 42.63 | 47.56 | 35.46 | 45.26 | 34.77 | 40.61 | 29.69 | 63.40 | 52.91 |
| ✔ UCIR (Hou et al., 2019) | 63.83 | 54.75 | 60.94 | 50.75 | 59.46 | 47.00 | 49.26 | 39.18 | 48.85 | 37.74 | 43.02 | 30.82 | 67.59 | 55.89 |
| ✔ RMM (Liu et al., 2021b) | 68.86 | 59.00 | 67.61 | 59.03 | – | – | – | – | – | – | – | – | 78.47 | 71.40 |
| ✘ EWC (Kirkpatrick et al., 2017) | 24.23 | 9.00 | 21.15 | 8.50 | 16.26 | 7.75 | 18.83 | 5.98 | 15.90 | 3.59 | 12.57 | 5.00 | 20.26 | 9.03 |
| ✘ LwF (Li & Hoiem, 2017) | 32.54 | 14.25 | 17.91 | 5.88 | 14.95 | 5.50 | 22.35 | 7.11 | 17.52 | 4.82 | 12.75 | 4.39 | 23.57 | 11.54 |
| ✘ LwF-MC (Li & Hoiem, 2017) | 46.06 | 33.38 | 27.31 | 15.75 | 19.99 | 11.88 | 29.09 | 15.46 | 23.22 | 13.23 | 17.46 | 8.16 | 31.22 | 20.69 |
| ✘ MUC (Liu et al., 2020b) | 49.56 | 36.00 | 32.35 | 20.63 | 22.68 | 9.50 | 32.59 | 19.18 | 26.83 | 15.28 | 22.08 | 10.41 | 35.03 | 24.46 |
| ✘ PASS (Zhu et al., 2021) | 63.88 | 55.75 | 60.07 | 49.13 | 58.21 | 48.75 | 49.88 | 41.86 | 47.30 | 39.38 | 42.04 | 32.86 | 62.26 | 50.63 |
| ✘ iVoro (Best)‡ | **83.57** | **74.40** | **83.52** | **74.39** | **81.24** | **71.45** | **81.74** | **72.34** | **80.22** | **71.13** | **79.08** | **69.95** | **90.04** | **83.84** |
| *imp.↑* | +19.69 | +18.65 | +23.45 | +25.26 | +23.03 | +22.70 | +31.86 | +30.48 | +32.92 | +31.75 | +37.04 | +37.09 | +27.78 | +33.21 |

for progressive VD construction (iVoro-D), (**III**) augmentation consensus/integration (iVoro-AC/AI), and (**IV**) multilayer VD (iVoro-L). In this section, our main goals are to: (1) validate the strength of every single component; (2) exhaust as many combinations of components as possible to see how different combinations collaboratively contribute to the overall result; and (3) investigate at which circumstances a method does or does not work, by analyzing data size, number of layers, and quantitative uncertainty.

**3.1 Datasets, Benchmarks, and Implementation Details.** Three standard datasets, CIFAR-100 (Krizhevsky et al., 2009), TinyImageNet (Le & Yang, 2015) and ImageNet-Subset (Deng et al., 2009a) for CIL are used for method evaluation. We follow the popular benchmarking protocol in exemplar-free CIL used by (Liu et al., 2021b; Zhu et al., 2021; Douillard et al., 2020; Hou et al., 2019) in which the inital phase contains a half of the classes while the subsequent phases each has $\frac{1}{5}$, $\frac{1}{10}$, or $\frac{1}{20}$ of the remaining classes. We mainly compare our method to non-exemplar methods including EWC (Kirkpatrick et al., 2017), LwF (Li & Hoiem, 2017), LwF-MC (Li & Hoiem, 2017), LwM (Dhar et al., 2019), and MUC (Liu et al., 2020b), but we also compare with several recent exemplar-based methods iCaRL (Rebuffi et al., 2017), EEIL (Castro et al., 2018), UCIR (Hou et al., 2019), and RMM (Liu et al., 2021b) for reference. A ResNet-18 (He et al., 2016) model is used for all experiments. We follow PASS (Zhu et al., 2021) to train the feature extractor on the first phase data but freeze it afterwards for all subsequent phases. All classes are expanded via rotating the original image by 90°, 180°, and 270°. See Appendix F for more details about the implementations of all the 12 ablation methods in Tab. 2.

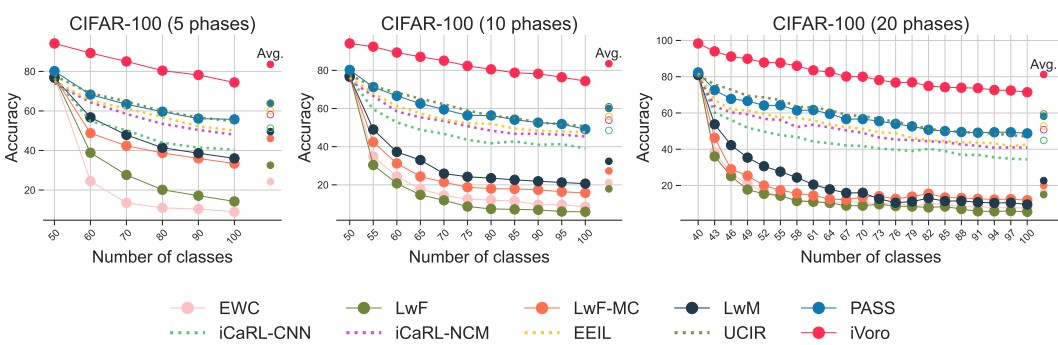

Figure 5: Top-1 classification accuracy on CIFAR-100 during 5/10/20 phases of CIL.

**3.2 iVoro: Simple VD is A Strong Baseline.** Surprisingly, by only using prototypes for VD construction, our baseline method iVoro can achieve competitive performance for short phases and much better results for long phases, compared to the state-of-the-art non-exemplar CIL method. For example, the difference in accuracy in comparison to PASS is 0.29%/6.91%/3.63% for 5/10/20-phase CIFAR-100, -3.58%/-1.09%/5.43% for 5/10/20-phase TinyImageNet, and 4.76% for ImageNet-Subset. We suspect that this is because the features generated by the frozen feature extractor can be

Table 2: Ablation experiments by testing with different combinations of parameterized feature normalization (★), progressive VD (♠), augmentation consensus (♣) or integration (♦), and multilayer VD (▼). See Table G.1 for the complete table.

| | | CIFAR-100 | | | TinyImageNet | | | ImageNet-Subset |
|---|---|---|---|---|---|---|---|---|
| | | 5 phases | 10 phases | 20 phases | 5 phases | 10 phases | 20 phases | 10 phases |
| | Methods | Avg. (Last) | Avg. (Last) | Avg. (Last) | Avg. (Last) | Avg. (Last) | Avg. (Last) | Avg. (Last) |
| ★♠♣♦▼ | iVoro | 66.39 (56.05) | 66.09 (56.05) | 62.33 (52.38) | 45.12 (38.27) | 45.09 (38.29) | 45.04 (38.29) | 66.50 (55.40) |
| ★♠♣♦▼ | iVoro-ND | 67.55 (57.25) | 66.89 (56.75) | 64.65 (54.72) | 51.83 (43.43) | 50.71 (42.48) | 50.17 (42.10) | 69.07 (58.52) |
| ★♠♣♦▼ | iVoro-AC | 81.38 (70.63) | 81.25 (70.63) | 78.16 (66.14) | 64.01 (55.26) | 64.01 (55.28) | 64.00 (55.29) | 83.41 (71.90) |
| ★♠♣♦▼ | iVoro-AI | 62.33 (50.18) | 60.37 (50.18) | 65.39 (59.11) | 55.60 (48.11) | 55.63 (48.12) | 55.50 (48.11) | 72.47 (60.66) |
| ★♠♣♦▼ | iVoro-NDAI | 69.00 (56.35) | 63.76 (52.87) | 63.94 (57.52) | 81.00 (71.29) | 79.64 (70.10) | 78.17 (68.70) | 86.92 (78.64) |
| ★♠♣♦▼ | iVoro-NDAC | 82.31 (72.04) | 82.29 (72.19) | 80.53 (70.01) | 59.75 (49.78) | 59.75 (49.80) | 59.74 (49.78) | 84.31 (73.72) |
| ★♠♣♦▼ | iVoro-NDACL | 83.57 (74.40) | 83.52 (74.39) | 72.64 (61.14) | 59.49 (50.09) | 59.51 (50.10) | 59.52 (50.13) | 84.83 (76.24) |
| ★♠♣♦▼ | iVoro-NDAIL | 77.57 (66.54) | 72.50 (62.28) | 81.24 (71.45) | 81.74 (72.34) | 80.22 (71.13) | 79.08 (69.95) | 90.04 (83.84) |

satisfactorily separable by linear bisectors (Fig. 2). As we can see, the features for other methods are all dramatically changing during the phases, but those for iVoro are all fixed, making incremental VD construction possible. Moreover, the accuracy of the last phase usually drops significantly with longer task sequence (e.g. 20 phases vs. 5 phases), but iVoro is highly robust at the last phase, because the final VDs are the same no matter how many phases it goes through. These results show that iVoro works favorably with long phases. When parameterized normalization is applied, iVoro-N further consistently improves upon iVoro by up to 2.40% (10-phase ImageNet-Subset) (see Tab. 2), by encouraging the compactness of feature distribution. See Appendix H about the detailed analysis of iVoro-N.

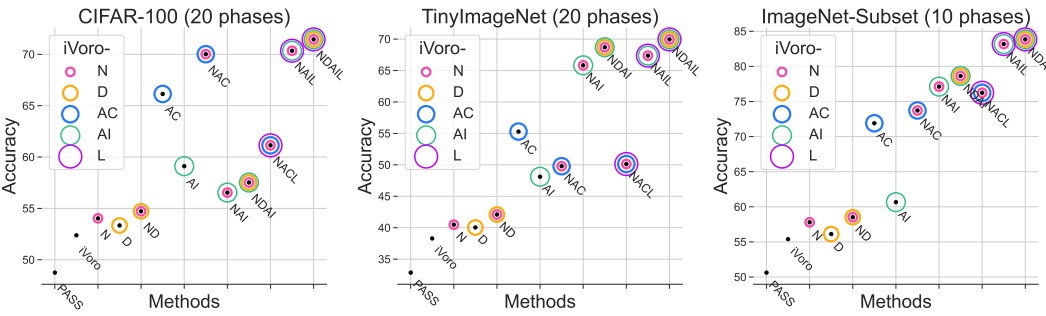

Figure 6: Illustration of the performance for the ablation methods shown in Table 2.

**3.3 Normalization (iVoro-N) and D&C (iVoro-D): Synergistic Effects.** Our very baseline method, iVoro, ignores at which phase a class was learned, and computes prototypes indifferently to construct the VD (i.e. 1-nearest neighbor model). To determine the decision boundaries more subtly, iVoro-D focuses on the refinement of the within-phase boundaries. These two components, iVoro-N/D, can individually improve iVoro, but also have collective impacts. For example, as shown in Tab. 2 and Fig. 6,

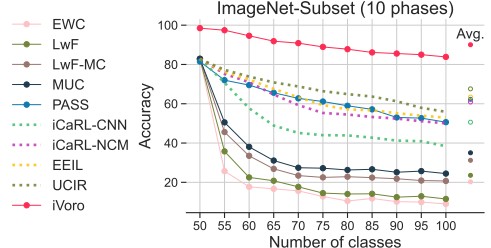

Figure 7: Top-1 classification accuracy on ImageNet-Subset (10 phases).

iVoro-ND > iVoro-N/iVoro-D > iVoro, corroborating that every single contribution is useful and necessary. More specifically, in the three datasets, iVoro-D makes the largest contribution to TinyImageNet (1.75%-3.44%) than to CIFAR-100 (0.48%-0.96%) or to ImageNet-Subset (0.72%). This can be explained by the fact that there are 100 classes in the first phase in the TinyImageNet dataset, while only 50 in the other two, making TinyImageNet a harder dataset if only vanilla prototypes are used to construct the VD.

**3.4 Why and When Will Augmentation Integration (iVoro-AC/AI) Help?** When augmentation consensus (iVoro-AC) or integration (iVoro-AI) is applied, the improvement is significant. For example, iVoro-AC obtains 13.76%, 17.00%, and 16.50% improvements upon iVoro on CIFAR-100,

TinyImageNet, and ImageNet-Subset, respectively. iVoro-AI itself is worse than iVoro-AC, but if combined with normalization and D&C, it further elevates the accuracy by a large margin, e.g. as high as 68.70% (iVoro-NDAI) on 20-phase TinyImageNet and 78.64% on 10-phase ImageNet-Subset. To investigate the reason of this prominent improvement, we calculate the entropy-based geometric variance in class level and plot them as a function of the Δaccuracy (i.e. the improvement in accuracy after augmentation integration is used), as shown in Fig. 8 and Fig. G.2. Interestingly, there is a clear correlation between HV and Δaccuracy, and this is more notable on ImageNet-Subset (Pearson's R ~0.9), probably because of its high resolution ($224 \times 224$). This tendency suggests that the higher the variance within the assignments from augmented images to expanded classes, the better the improvement after using augmentation integration. See Appendix I for uncertainty analysis, and Appendix N/Appendix O for class-level/sample-level analysis.

**3.5 How Good Should the Feature Extractor Be?**
As iVoro is heavily dependent on the feature extractor, which cannot be evolved in any way along the learning process, one may wonder if our method still work with a poorly trained feature extractor. To verify this, we gradually decrease the number of classes used to train the feature extractor $\phi$. As shown in Appendix J, compared with PASS, the best version of iVoro still has 17.75%, 13.59%, 10.89%, and 1.60% improvements with 40, 30, 20, and 10 initial classes, respectively. This means that, even if there is no strong feature extractor, our method can still reach acceptable performance higher than the state-of-the-art method.

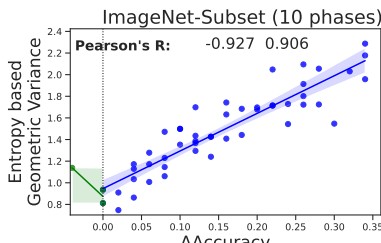

Figure 8: Entropy-based geometric variance (HV) in class level as a function of Δaccuracy (ImageNet-Subset).

**3.6 iVoro-L: VD Can Also Go Deeper.** In this section, we extract the feature from the 3$^{rd}$ block and establish a Cluster-to-cluster Voronoi Diagram (CCVD) using features from both layers. As expected, shown in Appendix K, the overall performance degenerates substantially, e.g. iVoro-NAI drops from 65.84% to 42.03%. However, and interestingly, if

Table 3: Comparison of iVoro to joint training in which all classes are learned in the same phase. The accuracy from ResNet-18 on the original label set are marked **bold**, while $\otimes$ denotes result upon the augmented label set ($4\times$). The results of iVoro on 5/10/20 phases of CIL are also shown. The best results are underlined.

| | Methods | CIFAR-100 | TinyImageNet | ImageNet-Subset |
|---|---|---|---|---|
| **Joint** | ResNet-18 | **75.34** | **57.37** | **80.44** |
| | ResNet-18$\otimes$ | 66.94 | 52.78 | 69.56 |
| | iVoro | 64.21 | 47.56 | 78.54 |
| | iVoro-AC | 81.65 | 67.99 | 92.72 |
| | iVoro-AI | 80.66 | 74.91 | 93.82 |
| **CIL** | iVoro (Best) | 74.40  74.39  71.45 | 72.34  71.13  69.95 | 83.84 |

integrated with CCVD and D&C, iVoro-NDAIL obtains even higher accuracy of 72.34%. The final results are presented in Tab. 1, Fig. K.3, and Appendix G. Our final model, multilayer VD, surpasses all previous methods by a large margin of 25.26%, 37.09%, and 33.21% on CIFAR-100, TinyImageNet, and ImageNet-Subset, respectively, even higher than exemplar-based CIL methods.

**3.7 Comparison with Joint Training.** In CIL, the classes are sequentially learned at each phase, whereas joint training simultaneously learns all the classes in the same phase, providing an upper bound for our CIL experiments. In Tab. 3, iVoro (and its variants) is applied to joint training, and is compared with ResNet on both the original and expanded label sets. Although there is still a substantial gap between iVoro (best) and the upper bound, the catastrophic forgetting is considerably overcome (see Appendix M). In addition, and surprisingly, iVoro-AC/AI can also promote the performance of joint training, e.g. +6.31%, +17.54%, and +13.38% for CIFAR-100, TinyImageNet, and ImageNet-Subset, respectively, suggesting that our augmentation integration method is also beneficial to general training where self-supervised label augmentation is involved.

# 4 CONCLUSION

In this paper, we use progressive Voronoi Diagram to model the class-incremental learning problem, and propose a number of new techniques that handle various aspects of this VD construction process that gradually and greatly improve the CIL performance. Thus, iVoro is shown to be a flexible, scalable, and robust framework that strictly maintains the privacy of previous data. Our code is available at https://machunwei.github.io/ivoro/.

ACKNOWLEDGMENTS

This research was supported in part by NSF through grant IIS-1910492.

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

# A    EXTENDED RELATED WORK

## A.1    RECENT PROGRESS IN INCREMENTAL LEARNING

Incremental Learning (Rebuffi et al., 2017; Hou et al., 2019; Wu et al., 2019; Zhu et al., 2021; Liu et al., 2021b) requires continuously updating a model using a sequence of new tasks without forgetting the old knowledge, which is also referred to as continual learning (Parisi et al., 2019; Delange et al., 2021; Chaudhry et al., 2019). The main challenge of incremental learning is catastrophic forgetting (McCloskey & Cohen, 1989; French, 1999; Goodfellow et al., 2014; Kemker et al., 2018), where deep neural network is prone to performance deterioration on the previously learned tasks as the model parameters overfit to the current data to optimize the stability-plasticity trade-off.

### A.1.1    CAUSATION OF CATASTROPHIC FORGETTING

Generally speaking, in deep neural networks, catastrophic forgetting (McCloskey & Cohen, 1989; Goodfellow et al., 2014; Kemker et al., 2018) comes from two sources: the feature distribution shifting of the old classes in the feature embedding space as well as the confusion and imbalance of the decision boundary of the classifier when learning new task. The former is caused by the excessive plasticity and parameter changing of the feature extractor of the deep model during finetuning on unseen data/classes, thus deteriorates the feature extraction and prediction on previous classes; while the latter is due to the highly overfitting and bias of the classifier on current task as well as the overlapping between the representation of new and old classes in the feature space.

### A.1.2    INCREMENTAL LEARNING SCENARIOS

Three common incremental learning scenarios are widely explored in recent papers (Van de Ven & Tolias, 2019).

*Task-incremental learning* (TIL) (Ostapenko et al., 2019; Shin et al., 2017; Kirkpatrick et al., 2017; Zenke et al., 2017; Wu et al., 2018; Lopez-Paz & Ranzato, 2017; Chaudhry et al., 2019; Buzzega et al., 2020; Cha et al., 2021; Pham et al., 2021; Fernando et al., 2017) incrementally learns a sequence of tasks in multiple phases, where each task contains unseen data of a new set of classes. To mitigate catastrophic forgetting, TIL assumes a simple setting where the task identity is known at inference time. The methods under this scenario keep leaning new task-independent classifiers or growing the model capacity by attaching additional modules (e.g. kernels, layers or branches), each corresponding to a specific task or a subset of classes. Since the task ID is available during inference, the model can directly select proper classifier or module without inferring task identity, which effectively solves the confusion boundary and classifier bias between old and new tasks, and often achieves satisfying performance. However, knowing task identity at test time is normally unrealistic in real-world situation hence restricts practical usage. Moreover, it may incur unbounded memory consumption for super long task sequence if increasing the model capacity for new tasks.

Unlike TIL constrained by the availability of task identity, *class-incremental learning* (CIL) (Liu et al., 2020b; Belouadah & Popescu, 2019; Chaudhry et al., 2018a; Zhu et al., 2021; Douillard et al., 2020; Rebuffi et al., 2017; Hou et al., 2019; Liu et al., 2020a; 2021a;b) updates a unified classifier for all classes learned so far while task identity is no longer required during inference. To compensate the missing task identity and alleviate forgetting issue, a branch of works (Rebuffi et al., 2017; Hou et al., 2019; Liu et al., 2021b; Douillard et al., 2020; Castro et al., 2018; Wu et al., 2019) alternatively follow a memory-based setting, in which a limited number of samples from old classes (e.g., 20 exemplars per class) is stored and maintained in a memory buffer, which are later replayed to jointly train the model with current data (normally combined with knowledge distillation) in order to constrain the feature distribution shifting of the old classes and the decision boundary bias of the classifier. However, their performance deteriorates with smaller buffer size, and eventually, the storing and sharing of previous data, e.g. medical images, may not be feasible when memory limits and privacy issue are taken into consideration. Given the potential memory issue, another direction of works (Kirkpatrick et al., 2017; Zenke et al., 2017; Li & Hoiem, 2017; Dhar et al., 2019; Zhu et al., 2021) intend to explore CIL in a much challenging setting without memory rehearsal, mainly based on regularization and knowledge distillation techniques, which is known as exemplar-free CIL. In this paper, we are following this CIL setting.

*Domain-incremental learning* (DIL) (Rostami, 2021; Tang et al., 2021; Volpi et al., 2021), different from the aforementioned two scenarios, incrementally learning new domains of the same classes in each phase. Some domain adaptation techniques, e.g. meta learning, data shifting, domain randomization, are implemented in DIL to increase the model robustness and generalizability to handle various domain distributions. Since this scenario is not quite related to this paper, no detailed discussion will be included.

### A.1.3 CATEGORIES OF INCREMENTAL LEARNING METHODS

There are three categories of existing IL methods to overcome catastrophic forgetting (Delange et al., 2021).

*Regularization-based* methods constrain the plasticity of the model to preserve old knowledge. This can be addressed by directly penalizing the changes of important parameters for previous tasks (Aljundi et al., 2018; Chaudhry et al., 2018a; Kirkpatrick et al., 2017; Zenke et al., 2017; Kumar et al., 2021) or regularizing the gradients when training on unseen data (Lopez-Paz & Ranzato, 2017; Chaudhry et al., 2018b). Knowledge distillation is another regularization solution, which is widely used in various IL methods to implicitly consolidate previous knowledge by introducing regularization loss term on model representations, including output logits or probabilities (Li & Hoiem, 2017; Schwarz et al., 2018; Rebuffi et al., 2017; Castro et al., 2018) and intermediate features (Hou et al., 2019; Dhar et al., 2019; Douillard et al., 2020; Zhu et al., 2021). Some other works focus on correcting the classifier bias on new classes (Belouadah & Popescu, 2019; Wu et al., 2019; Belouadah & Popescu, 2020; Zhao et al., 2020).

*Rehearsal-based* methods either store and replay a limited amount of exemplars from old classes as raw images (Rebuffi et al., 2017; Hou et al., 2019; Liu et al., 2021b; Chaudhry et al., 2019; Buzzega et al., 2020) or embedded features (Hayes et al., 2020; Iscen et al., 2020) to jointly train the model in the incremental phases, or alternatively generate exemplars of previous classes (Ostapenko et al., 2019; Shin et al., 2017; Wu et al., 2018; Kemker & Kanan, 2017). The former relies on memory buffer for all learned classes, where the performance is constrained by the buffer size limits and it is impracticable when data privacy is required and storing data is prohibited. The latter requires continuously learning a deep generative model, which is also prone to catastrophic forgetting thus the quality of generated exemplars is not reliable.

*Architecture-based* methods aims at dynamically adapting task-specific sub-network architectures, which requires task identity to select proper sub-network. Some works directly expand the network by adding new layers or branches (Rusu et al., 2016; Li et al., 2019; Wang et al., 2017; Yoon et al., 2017), which is limited in practice due to unbounded model parameter growth. Others freeze partial network with masks for old tasks (Golkar et al., 2019; Hung et al., 2019; Mallya & Lazebnik, 2018; Serra et al., 2018), but suffering from running out of model parameters for new knowledge. The architecture-based methods are usually combined with memory buffer and distillation, and can achieve good results.

Our work is focusing on the most challenging but also the most practical non-exemplar class-incremental learning problem, which is a general real-world scenario when no old data can be stored due to memory limits or data privacy and task identity is unavailable during inference, with the constraint of fixed model capacity in the same time.

### A.2 COMPUTATIONAL GEOMETRY FOR DEEP LEARNING

Computational geometry is an emerging perspective for studying various aspects of deep learning. The geometric structure of deep neural networks is first hinted at by (Raghu et al., 2017) which reveals that piecewise linear activations subdivide input space into convex polytopes. Afterward, (Balestriero et al., 2019) points out that the exact structure is a Power Diagram (PD, a generalized form of Voronoi Diagram) (Aurenhammer, 1987) which is subsequently used to explain the recurrent neural networks (Wang et al., 2018) and generative models (Balestriero et al., 2020). The Power Diagram (or Voronoi Diagram) subdivision, however, is not necessarily the optimal model for describing the partitioning of deep/intermediate feature spaces. More recently, several works in computational geometry (Chen et al., 2013; 2017; Huang et al., 2021) use an influence function $F(\mathcal{C}, z)$ to measure the joint influence of all objects in $\mathcal{C}$ on a query $z$ to build a Cluster-induced Voronoi Diagram (CIVD), providing an advanced reform of the classical Voronoi Diagram. Observing that Prototypical

Network (Snell et al., 2017), a widely adopted metric-based few-shot learning (FSL) method, is essentially a Voronoi Diagram in the feature space, DeepVoro (Ma et al., 2022b) first unifies various kinds of FSL methods, and then constructs a CIVD by incorporating heterogeneous features, achieving the state-of-the-art performance in FSL.

Besides FSL, Voronoi Diagram subdivision has also been used for deep learning uncertainty calibration (Ma et al., 2021), adversarial robustness (Sitawarin et al., 2021), topological data analysis (Polianskii & Pokorny, 2019; 2020; Poklukar et al., 2022), and medical applications (Ma et al., 2018; 2019). In this paper, distinct from the three aforementioned lines of research (i.e. regularization-based, rehearsal-based, and architecture-based methods), we propose the geometry-based CIL method iVoro (and its variants), inspired by Voronoi Diagram subdivision.

## B    DEMONSTRATIVE ILLUSTRATION ON MNIST DATASET IN 2D SPACE

In Figure2, MNIST (LeCun, 1998), a small and simple dataset, was used for the illustration of four methods, fine-tuning, PASS (Zhu et al., 2021), iVoro, and iVoro-AC, because of the convenience of embedding the examples into $\mathbb{R}^2$. The total 10 classes are split into a sequence of 4, 3, and 3 classes. A ResNet-18 model is used as the feature extractor for all four methods. (**A**) In fine-tuning, the model is firstly trained on the 4 classes in the first phase, and then fine-tuned only on the subsequent 3 and 3 classes in phase 2 and phase 3. (**B**) In PASS, SSL-based label augmentation is applied on all three phases and expands the classes to be 16, 9, and 9 classes. The default hyper-parameters are used to train PASS (i.e. the weight for knowledge distillation is 10 and the weight for prototype augmentation is 10). To ensure the final subdivision of space is a Voronoi diagram, Thm. 2.1 (i.e. Voronoi diagram reduction in Algorithm 1) is applied during the training of fine-tuning and PASS. (**C**) In iVoro, the feature extractor from the first phase of (B) is frozen and used without fine-tuning for all the subsequent phases. The feature means are calculated as prototypes and no feature transformation is used. Note that only the features from the original images without rotation are used in iVoro. (**D**) The only difference with (C) is that all the expanded classes are also considered as independent cells, allowing for further integration.

**Result Analysis.** For (A) fine-tuning and (B) PASS, the model's accuracy for data at individual phases are also shown in shadow. Fine-tuning are able to achieve near-perfect prediction for classes in the current phase locally, but fails to maintain satisfactory performance on any historical class (accuracy $\sim 0\%$). PASS, on the other hand, basically deteriorates slightly on the classes from the first phase, due to the high weights on the KD loss and the prototype loss, but it also becomes almost incapable of learning on new classes (accuracy $\sim 0\%$). iVoro, i.e. the simplest 1-nearest-neighbor model, surprisingly obtains superior accuracy (64.84%, 24.28% higher than PASS) by only using a fixed feature extractor trained from only 4 classes (16 expanded classes). iVoro-AC achieves comparable result (61.77%) with iVoro, but the 2D embedding makes it harder to demonstrate the efficacy of our proposed method. From this 2D illustration, it is obvious that the much higher performance of iVoro/iVoro-AC is achieved through a much better space partitioning.

## C    NOTATIONS AND ACRONYMS

In this section, we list all the notations used in the Methodology in Tab. C.1, the notations and acronyms for various geometric structures used in the paper in Tab. C.2, and all ablation methods in Tab. C.3.

Table C.1: Complete list of all notations used in Methodology 2.

| Notations | Descriptions |
|---|---|
| $T$ | total phase, $T \in \mathbb{R}$ |
| $t$ | current phase, $t \in \{1, ..., T\}$ |
| $\tau$ | historical phase, $\tau \in \{1, ..., t\}$ |
| $\mathcal{D}_t$ | dataset in phase $t$ |
| $(\boldsymbol{x}_{t,i}, y_{t,i})$ | data (image) and label in phase $t$, $i \in \{1, ..., N_t\}, t \in \{1, ..., T\}$ |
| $\mathcal{C}_t$ | the set of classes in phase $t$ |
| $\mathcal{C}_{t,k}$ | the $k^{\text{th}}$ class in phase $t$ |
| $N_{t,k}$ | number of examples in class $k$ at phase $t$ |
| $N_t$ | number of all examples in phase $t$, i.e. $N_t = \sum_{k=1}^{K_t} N_{t,k}$ |
| $\phi$ | feature extractor (typically a deep neural network) |
| $\phi^{(l)}$ | feature extractor, but only outputs the features from the $l^{\text{th}}$ layer |
| $\boldsymbol{z}$ | feature for $\boldsymbol{x}$, i.e. $\boldsymbol{z} = \phi(\boldsymbol{x}), \boldsymbol{z} \in \mathbb{R}^n$ |
| $\theta$ | classification head, can be either a Voronoi Diagram, or logistic regression |
| $\boldsymbol{c}_{\tau,k}$ | prototypical Voronoi center for phase $\tau$ and class $\mathcal{C}_{\tau,k}$ |
| $\tilde{\boldsymbol{c}}_{\tau,k}$ | linear probing-induced Voronoi center for phase $\tau$ and class $\mathcal{C}_{\tau,k}$ |
| $f$ | $L_2$ normalization |
| $g$ | linear transformation $g_{w,\eta}(\boldsymbol{z}) = w\boldsymbol{z} + \eta$ |
| $h$ | Tukey's ladder of powers transformation, parameterized by $\lambda$ |
| $\boldsymbol{W}_{t,k}, b_{t,k}$ | linear weight and bias for class $\mathcal{C}_{t,k}$ |
| $\boldsymbol{v}, q$ | parameters for the linear bisector $\boldsymbol{v}^T \boldsymbol{z}' - q = 0$ |
| $\alpha$ | index of four rotations, $\alpha \in \{0, 1, 2, 3\}$ |
| $\boldsymbol{d}^{(\alpha,\alpha')} \in \mathbb{R}^K$ | the collection of the distances from $\phi(\boldsymbol{x}^{\alpha'})$, to $K$ classes with rotation index $\alpha'$ |
| HV | Entropy-based geometric variance |

Table C.2: Notations and acronyms for VD, PD, and CCVD, three geometric structures used in the paper.

| Geometric Structures | Acronyms | Notations | Description |
|---|---|---|---|
| Voronoi Diagram | VD | $\boldsymbol{c}_k$ | center for a Voronoi cell $\omega_k, k \in \{1, .., K\}$ |
| | | $\omega_k$ | dominating region for center $\boldsymbol{c}_k, k \in \{1, .., K\}$ |
| Power Diagram (Aurenhammer, 1987) | PD | $\boldsymbol{c}_k$ | center for a Power cell $\omega_k, k \in \{1, .., K\}$ |
| | | $\nu_k$ | weight for center $\boldsymbol{c}_k, k \in \{1, .., K\}$ |
| | | $\omega_k$ | dominating region for center $\boldsymbol{c}_k, k \in \{1, .., K\}$ |
| Cluster-to-cluster Voronoi Diagram (Ma et al., 2022a) | CCVD | $\mathcal{C}_k$ | cluster as the "center" for a CCVD cell $\omega_k, k \in \{1, .., K\}$ |
| | | $\omega_k$ | dominating region for cluster $\mathcal{C}_k$ |
| | | $\mathcal{C}(\boldsymbol{z})$ | the cluster that query point $\boldsymbol{z}$ belongs |
| | | $F$ | influence function $F(\mathcal{C}_k, \mathcal{C}(\boldsymbol{z}))$ from $\mathcal{C}_k$ to query cluster $\mathcal{C}(\boldsymbol{z})$ |
| | | $\alpha$ | magnitude of the influence |

Table C.3: Complete list of all variants of iVoro.

| Methods | Prototype | Normalization | D&C | Augmentation Consensus | Augmentation Integration | Multilayer VD |
|---|---|---|---|---|---|---|
| iVoro | ✔ | ✘ | ✘ | ✘ | ✘ | ✘ |
| iVoro-N | ✔ | ✔ | ✘ | ✘ | ✘ | ✘ |
| iVoro-D | ✔ | ✘ | ✔ | ✘ | ✘ | ✘ |
| iVoro-ND | ✔ | ✔ | ✔ | ✘ | ✘ | ✘ |
| iVoro-AC | ✔ | ✘ | ✘ | ✔ | ✘ | ✘ |
| iVoro-AI | ✔ | ✘ | ✘ | ✘ | ✔ | ✘ |
| iVoro-NAC | ✔ | ✔ | ✘ | ✔ | ✘ | ✘ |
| iVoro-NAI | ✔ | ✔ | ✘ | ✘ | ✔ | ✘ |
| iVoro-NDAC | ✔ | ✔ | ✔ | ✔ | ✘ | ✘ |
| iVoro-NDAI | ✔ | ✔ | ✔ | ✘ | ✔ | ✘ |
| iVoro-NACL | ✔ | ✔ | ✘ | ✔ | ✘ | ✔ |
| iVoro-NAIL | ✔ | ✔ | ✘ | ✘ | ✔ | ✔ |
| iVoro-NDACL | ✔ | ✔ | ✔ | ✔ | ✘ | ✔ |
| iVoro-NDAIL | ✔ | ✔ | ✔ | ✘ | ✔ | ✔ |

# D    Dataset Details

Here we give the detailed statistics of the three datasets used in the paper. Augmentation consensus (iVoro-AC) and augmentation integration (iVoro-AI) work more favorably with images with higher resolution, e.g. ImageNet-Subset (see Fig. 8), as the rotation operation makes less sense if the image is too blur.

Table D.1: Summarization of the datasets used in the paper.

| Datasets | Image size | Training Images | Total Classes | Number of Phases |
|---|---|---|---|---|
| CIFAR-100 (Krizhevsky et al., 2009) | $32 \times 32 \times 3$ | 60000 | 100 | 5, 10, 20 |
| TinyImageNet (Le & Yang, 2015) | $64 \times 64 \times 3$ | 100000 | 200 | 5, 10, 20 |
| ImageNet-Subset (Deng et al., 2009b) | $224 \times 224 \times 3$ | 130000 | 100 | 10 |

# E    Power Diagram Subdivision and Voronoi Reduction

In this section we provide the proof of Theorem 2.1.

**Lemma E.1.**  The vertical projection from the lower envelope of the hyperplanes $\{\Pi_k(\boldsymbol{z}) : \boldsymbol{W}_k^T \boldsymbol{z} + b_k\}_{k=1}^K$ onto the input space $\mathbb{R}^n$ defines the cells of a PD.

**Theorem 2.1** (Voronoi Diagram Reduction (Ma et al., 2022a)). The linear classifier parameterized by $\boldsymbol{W}, \boldsymbol{b}$ partitions the input space $\mathbb{R}^n$ to a Voronoi Diagram with centers $\{\tilde{\boldsymbol{c}}_1, ..., \tilde{\boldsymbol{c}}_K\}$ given by $\tilde{\boldsymbol{c}}_k = \frac{1}{2}\boldsymbol{W}_k$ if $b_k = -\frac{1}{4}||\boldsymbol{W}_k||_2^2, k = 1, ..., K$.

*Proof.*  We first articulate Lemma E.1 and find the exact relationship between the hyperplane $\Pi_k(\boldsymbol{z})$ and the center of its associated cell in $\mathbb{R}^n$. By Definition 2.1, the cell for a point $\boldsymbol{z} \in \mathbb{R}^n$ is found by comparing $d(\boldsymbol{z}, \boldsymbol{c}_k)^2 - \nu_k$ for different $k$, so we define the power function $p(\boldsymbol{z}, S)$ expressing this value

$$p(\boldsymbol{z}, S) = (\boldsymbol{z} - \boldsymbol{u})^2 - r^2 \tag{2}$$

in which $S \subseteq \mathbb{R}^n$ is a sphere with center $\boldsymbol{u}$ and radius $r$. In fact, the weight $\nu$ associated with a center in Definition 2.1 can be interpreted as the square of the radius $r^2$. Next, let $U$ denote a paraboloid $y = \boldsymbol{z}^2$, let $\Pi(S)$ be the transform that maps sphere $S$ with center $\boldsymbol{u}$ and radius $r$ into hyperplane

$$\Pi(S) : y = 2\boldsymbol{z} \cdot \boldsymbol{u} - \boldsymbol{u} \cdot \boldsymbol{u} + r^2. \tag{3}$$

It can be proved that $\Pi$ is a bijective mapping between arbitrary spheres in $\mathbb{R}^n$ and nonvertical hyperplanes in $\mathbb{R}^{n+1}$ that intersect $U$ (Aurenhammer, 1987). Further, let $\boldsymbol{z}'$ denote the vertical projection of $\boldsymbol{z}$ onto $U$ and $\boldsymbol{z}''$ denote its vertical projection onto $\Pi(S)$, then the power function can be written as

$$p(\boldsymbol{z}, S) = d(\boldsymbol{z}, \boldsymbol{z}') - d(\boldsymbol{z}, \boldsymbol{z}''), \tag{4}$$

which implies the following relationships between a sphere in $\mathbb{R}^n$ and an associated hyperplane in $\mathbb{R}^{n+1}$ (Lemma 4 in (Aurenhammer, 1987)): let $S_1$ and $S_2$ be nonco-centeric spheres in $\mathbb{R}^n$, then the bisector of their Power cells is the vertical projection of $\Pi(S_1) \cap \Pi(S_2)$ onto $\mathbb{R}^n$. Now, we have a direct relationship between sphere $S$, and hyperplane $\Pi(S)$, and comparing equation (3) with the hyperplanes used in logistic regression $\{\Pi_k(\boldsymbol{z}) : \boldsymbol{W}_k^T \boldsymbol{z} + b_k\}_{k=1}^K$ gives us

$$\boldsymbol{u} = \frac{1}{2}\boldsymbol{W}_k$$
$$r^2 = b_k + \frac{1}{4}||\boldsymbol{W}_k||_2^2. \tag{5}$$

Although there is no guarantee that $b_k + \frac{1}{4}||\boldsymbol{W}_k||_2^2$ is always positive for an arbitrary logistic regression model, we can impose a constraint on $r^2$ to keep it be zero during the optimization, which implies

$$b_k = -\frac{1}{4}||\boldsymbol{W}_k||_2^2. \tag{6}$$

By this way, the radii for all $K$ spheres become identical (all zero). After the optimization of logistic regression model, the centers $\{\frac{1}{2}\boldsymbol{W}_k\}_{k=1}^K$ will be used as probing-induced Voronoi centers.    □

## F  IMPLEMENTATION DETAILS AND RESULT ANALYSIS OF COMPREHENSIVE ABLATION STUDIES

**iVoro.** We generally follow the protocol of PASS (Zhu et al., 2021) to train the feature extractor $\phi$ but *only* on the data from the first phase, i.e. 50 (for 5/10 phases) or 40 (for 20 phases) classes of CIFAR-100, 100 classes of TinyImageNet, and 50 classes of ImageNet-Subset. We also reproduce the results of PASS, using the same hyper-parameters, e.g. the weight for the knowledge distillation loss set at 10, and the weight for the prototype augmentation loss set at 10. For iVoro (and all its subsequent variants), the trained model is frozen after the first phase and throughout all the remaining phases. At phase $t$, the prototypical centers $\{c\}$ are computed for *both the current and the historical phases* $\tau \in \{1, ..., t\}$, and are used to construct the Voronoi Diagram.

The simplest iVoro method (i.e. the vanilla Voronoi Diagram, or 1-nearest-neighbor) can already achieve comparable or even better results than the state-of-the-art non-exemplar CIL methods. For example, the difference in accuracy in comparison to PASS is 0.29%/6.91%/3.63% for 5/10/20-phase CIFAR-100, -3.58%/-1.09%/5.43% for 5/10/20-phase TinyImageNet, and 4.76% for 10-phase ImageNet-Subset. Notably, there is always a significant elevation of accuracy on long-phase data, suggesting the continuous fine-tuning of model, even with improved loss functions, tends to forget seriously on earlier data. With a fixed feature extractor, iVoro has shown an improved ability to overcome catastrophic forgetting. On the other hand, for short-phase data, iVoro is similar or worse than the state-of-the-art method, probably because of the prototypical centers are computed without considering data distribution (i.e. simply the mean of features).

**iVoro-N.** To inspect the effectiveness of the parameterized feature transformation, we apply $L_2$ normalization with/without Tukey's ladder of powers transformation ($\lambda$ varying from 0.3 to 0.9), and compare with iVoro. Generally, the improvement acquired from the feature normalization is more prominent on more complex datasets (e.g. TinyImageNet and ImageNet-Subset), or simpler datasets with longer phases (e.g. CIFAR-100 with 20 phases), with improvements ranging from 1.65% to 2.40% higher than iVoro. The detailed analysis is presented in Sec. H.

**iVoro-D/iVoro-ND.** The detailed algorithm of iVoro-D is presented in Alg. 3. Specifically, for each phase $\tau \in \{1, ..., t\}$, the local dataset $\mathcal{D}_\tau$ is used to train a logistic regression model (restricted by Thm. 2.1) with weight decay $\beta$ at 0.0001 and initial learning rate at 0.001. The result is also shown in Tab. 2. Aided by the D&C algorithm and local logistic regression, iVoro-D is consistently better than iVoro, e.g. 0.48%~0.96% higher on the CIFAR-100 dataset, 1.75%~3.44% higher on the TinyImageNet dataset, and 0.72% higher on the ImageNet-Subset dataset. When further combined with feature normalization, iVoro-ND achieves even higher accuracy, 54.72% on 20-phase CIFAR-100, 42.10% on 20-phase TinyImageNet, and 58.52% on ImageNet-Subset. For comparison, PASS reaches 48.75% on 20-phase CIFAR-100, 32.86% on 20-phase TinyImageNet, and 50.63% on ImageNet-Subset. Therefore without incorporating more sophisticate techniques like iVoro-R/iVoro-AC/iVoro-L, the iVoro-ND method can already surpass previous state-of-the-art method by a large margin e.g. 5.97%/9.24%/7.89%.

**iVoro-AC/iVoro-AI/iVoro-NAC/iVoro-NAI.** While the previous variants of iVoro only consider the prediction on the original image/class, here we show that the prediction can be substantially improved by augmentation consensus (iVoro-AC) and augmentation integration (iVoro-AI) proposed in this paper. Specifically, iVoro-AC improves upon iVoro by 13.76%~14.58% on CIFAR-100, by 16.99%~17.00% on TinyImageNet, and by 16.50% on ImageNet-Subset. iVoro-AI itself generally works worse than iVoro-AC, e.g. improves up to 6.73% on CIFAR-100, up to 9.82% on TinyImageNet, and 5.26% on ImageNet-Subset, but if combined with feature normalization, iVoro-NAI performs much better than iVoro-NAC on TinyImageNet (up to 65.83%) and ImageNet-Subset (77.12%). When augmentation consensus/integration is used, the previous variants iVoro-N and iVoro-D can all be promoted. Generally, adding an additional component will bring in more performance gain, as shown in Tab. 2 in detail.

**iVoro-NACL/iVoro-NAIL/iVoro-NDAIL.** We further validate the multilayer VD for multiple feature spaces. As a proof of concept, we only extract the feature from the third block to build an additional VD and conduct the integration using CCVD (Def. 2.3) with $\gamma$ set at 1. Compared with iVoro-NAC, multilayer VD (iVoro-NACL) further improves both average accuracy and last accuracy on CIFAR-100 under 5/10 phases settings (2.36%/2.20% better on last phase, respectively), which also realizes the highest performance on the given settings among all ablation settings. The final

performance on ImageNet-Subset is also improved by 2.52%. In the meanwhile, multilayer VD does not make obvious difference on TinyImageNet. When adding multilayer VD to iVoro-NAI, labelled as iVoro-NAIL, significant performance gain is observed on all experiments of CIFAR-100 (9.1%/9.1%/15.76% average accuracy increments on 5/10/20 phases) and large improvement is also achieved on ImageNet-Subset (6.06% final accuracy growth). On the contrary, only limited gain is presented on TinyImageNet. The above ablation results may demonstrate that multilayer VD with augmentation integration works extraordinarily well for small class set (100 classes), but not very effective when class number increases.

**Robustness Analysis.** We run PASS 5 times on CIFAR-100 with the 10-phase setting, and the last accuracies (%) are 48.25%, 49.03%, 53.03%, 53.95%, and 54.75%, respectively, with mean and standard deviation (std) being 51.80%±2.65%; meanwhile, we also test PASS for 5 runs on ImageNet-Subset with 10 phases, and the last accuracies (%) are 49.85%, 50.63%, 51.03%, 51.88%, and 52.52%, respectively, with mean±std being 51.18%±0.94%. As shown above, even though PASS achieves relatively good accuracy on average, the training is not very stable on CIFAR-100, where the difference between the highest and lowest accuracy on CIFAR-100 is as large as 6.5%; while the performance on ImageNet-Subset is slightly better regarding the robustness, PASS still ranges from 49.85% to 52.52%. On the contrary, compared to PASS, our iVoro method is naturally robust to various datasets with no fluctuation in performance, due to the frozen feature extractor and unbiased classifier based on VD.

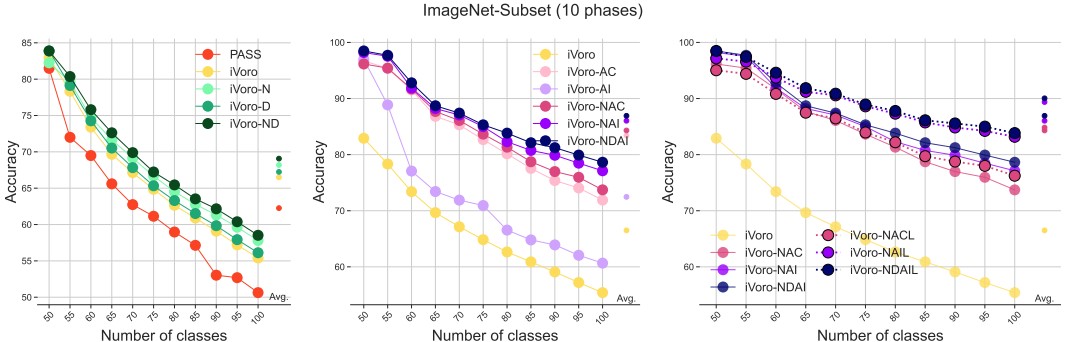

Figure F.1: Top-1 classification accuracy on ImageNet-Subset with all 12 ablation methods during 10 phases of CIL.

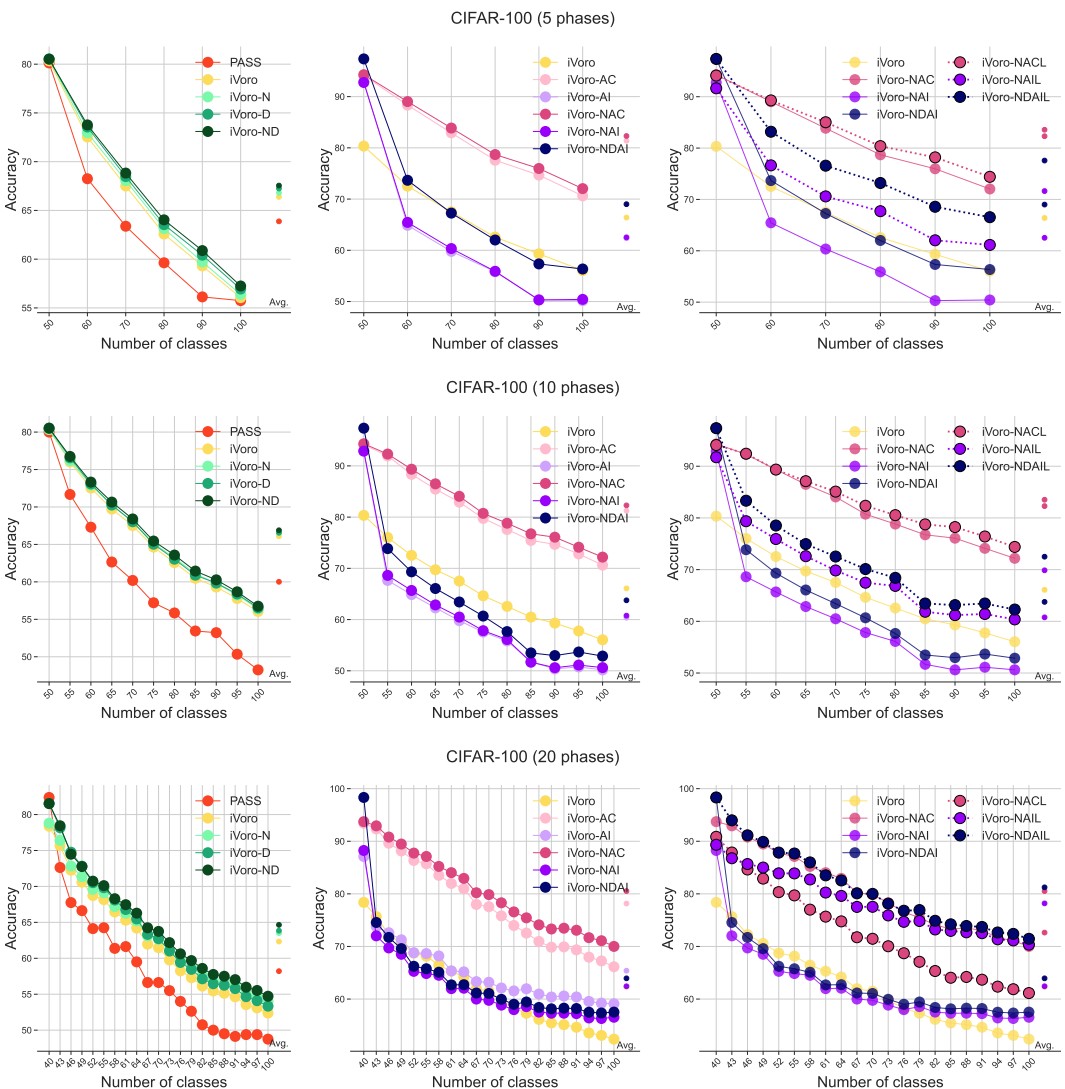

Figure F.2: Top-1 classification accuracy on CIFAR-100 with all 12 ablation methods during 5/10/20 phases of CIL.

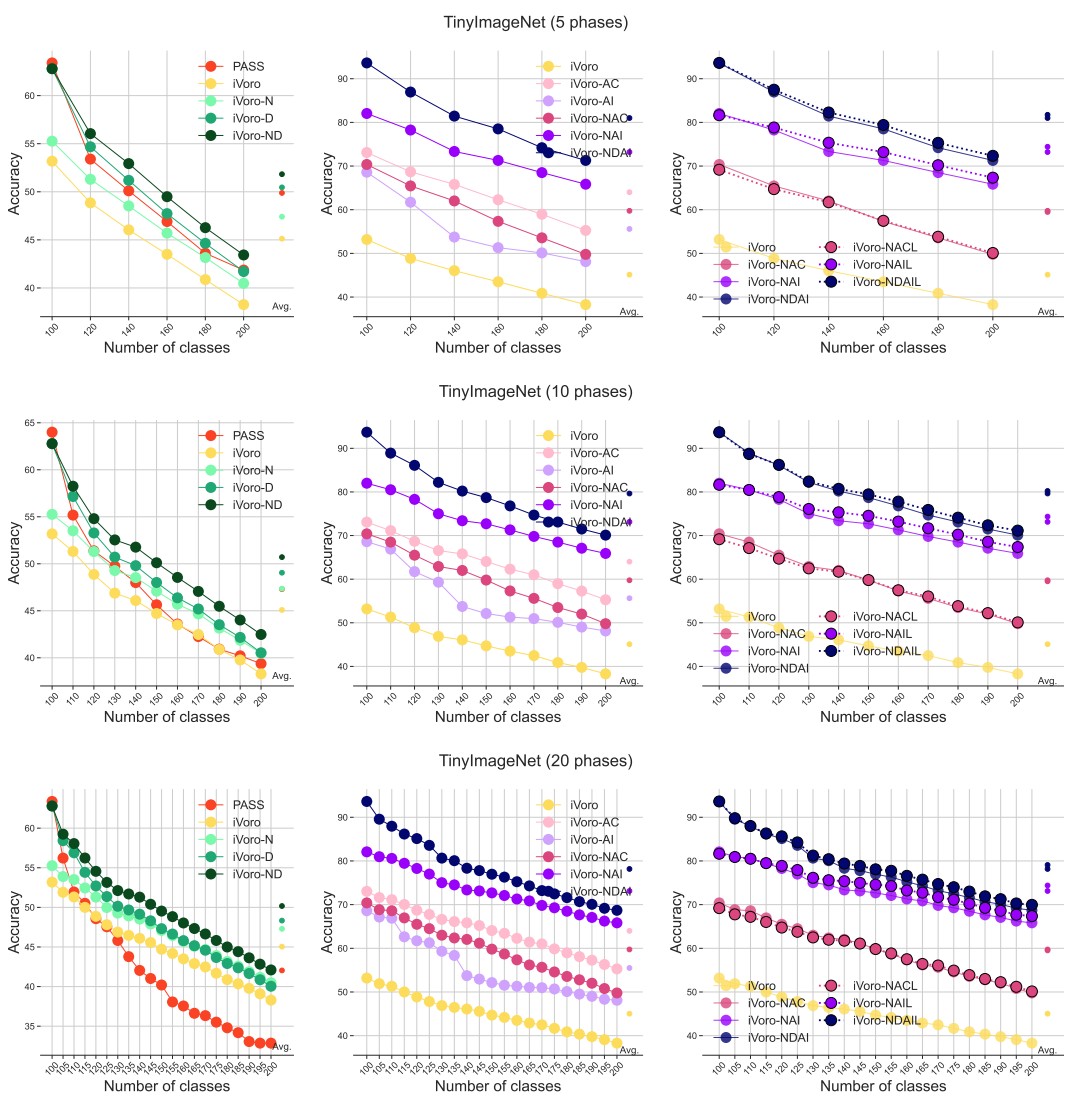

Figure F.3: Top-1 classification accuracy on TinyImageNet with all 12 ablation methods during 5/10/20 phases of CIL.

# G    ADDITIONAL FIGURES AND TABLES

Table G.1: Comprehensive ablation experiments by testing with different combinations of the five components: parameterized feature normalization (★), progressive VD via D&C (♠), augmentation consensus (♣) or integration (♦), and multilayer VD for multiple feature spaces (▼). The solid symbol (★♠♣♦▼) indicates a corresponding component is applied while the grayed symbol (★♠♣♦▼) means the component is ablated. To clearly show the improvement contributed to by a certain component, the colored numbers represent the relative improvements compared to a certain row indicated by the triangle (▶▶▶▶▶) with the same color. The red, blue, or green background indicates when Voronoi Diagram subdivision, augmentation consensus/integration, or multilayer Voronoi Diagram is introduced, respectively.

| | CIFAR-100 | | | | | | TinyImageNet | | | | | | ImageNet-Subset | |
|---|---|---|---|---|---|---|---|---|---|---|---|---|---|---|
| | 5 phases | | 10 phases | | 20 phases | | 5 phases | | 10 phases | | 20 phases | | 10 phases | |
| Methods | Avg. | Last | Avg. | Last | Avg. | Last | Avg. | Last | Avg. | Last | Avg. | Last | Avg. | Last |
| ▶ iVoro | 66.39 | 56.05 | 66.09 | 56.05 | 62.33 | 52.38 | 45.12 | 38.27 | 45.09 | 38.29 | 45.04 | 38.29 | 66.50 | 55.40 |
| ▶ iVoro-N ★♠♣♦▼ | 66.80 +0.42 | 56.39 +0.34 | 66.51 +0.43 | 56.40 +0.35 | 63.50 +1.17 | 54.03 +1.65 | 47.40 +2.29 | 40.48 +2.21 | 47.35 +2.26 | 40.48 +2.19 | 47.29 +2.25 | 40.48 +2.19 | 68.19 +1.69 | 57.80 +2.40 |
| iVoro-D ★♠♣♦▼ | 67.24 +0.85 | 56.94 +0.89 | 66.56 +0.47 | 56.53 +0.48 | 63.84 +1.51 | 53.34 +0.96 | 50.46 +5.34 | 41.71 +3.44 | 49.05 +3.96 | 40.53 +2.24 | 48.33 +3.28 | 40.04 +1.75 | 67.24 +0.74 | 56.12 +0.72 |
| iVoro-ND ★♠♣♦▼ | 67.55 +0.74 | 57.25 +0.86 | 66.89 +0.38 | 56.75 +0.35 | 64.65 +1.15 | 54.72 +0.69 | 51.83 +4.42 | 43.43 +2.95 | 50.71 +3.36 | 42.48 +2.00 | 50.17 +2.88 | 42.10 +1.62 | 69.07 +0.88 | 58.52 +0.72 |
| ▶ iVoro-AC ★♠♣♦▼ | 81.38 +15.00 | 70.63 +14.58 | 81.25 +15.16 | 70.63 +14.58 | 78.16 +15.84 | 66.14 +13.76 | 64.01 +18.90 | 55.26 +16.99 | 64.01 +18.92 | 55.28 +16.99 | 64.00 +18.96 | 55.29 +17.00 | 83.41 +16.90 | 71.90 +16.50 |
| ▶ iVoro-AI ★♠♣♦▼ | 62.33 -4.05 | 50.18 -5.87 | 60.37 -5.72 | 50.18 -5.87 | 65.39 +3.07 | 59.11 +6.73 | 55.60 +10.48 | 48.11 +9.84 | 55.63 +10.54 | 48.12 +9.83 | 55.50 +10.46 | 48.11 +9.82 | 72.47 +5.97 | 60.66 +5.26 |
| ▶ iVoro-NAC ★♠♣♦▼ | 82.31 +0.93 | 72.04 +1.41 | 82.29 +1.04 | 72.19 +1.56 | 80.53 +2.36 | 70.01 +3.87 | 59.75 -4.26 | 49.78 -5.48 | 59.75 -4.26 | 49.80 -5.48 | 59.74 -4.26 | 49.78 -5.51 | 84.31 +0.90 | 73.72 +1.82 |
| ▶ iVoro-NAI ★♠♣♦▼ | 62.53 +0.19 | 50.42 +0.24 | 60.77 +0.41 | 50.62 +0.44 | 62.43 -2.97 | 56.54 -2.57 | 73.21 +17.61 | 65.84 +17.73 | 73.14 +17.51 | 65.90 +17.78 | 73.13 +17.63 | 65.83 +17.72 | 86.04 +13.57 | 77.12 +16.46 |
| iVoro-NDAI ★♠♣♦▼ | 69.00 +6.47 | 56.35 +5.93 | 63.76 +2.99 | 52.87 +2.25 | 63.94 +1.51 | 57.52 +0.98 | 81.00 +7.79 | 71.29 +5.45 | 79.64 +6.50 | 70.10 +4.20 | 78.17 +5.04 | 68.70 +2.87 | 86.92 +0.89 | 78.64 +1.52 |
| iVoro-NACL ★♠♣♦▼ | **83.57** +1.25 | **74.40** +2.36 | **83.52** +1.23 | **74.39** +2.20 | 72.64 -7.89 | 61.14 -8.87 | 59.49 -0.26 | 50.09 +0.31 | 59.51 -0.24 | 50.10 +0.30 | 59.52 -0.22 | 50.13 +0.35 | 84.83 +0.51 | 76.24 +2.52 |
| ▶ iVoro-NAIL ★♠♣♦▼ | 71.63 +9.10 | 61.14 +10.72 | 69.87 +9.10 | 60.37 +9.75 | 78.19 +15.76 | 70.35 +13.81 | 74.42 +1.21 | 67.34 +1.50 | 74.35 +1.22 | 67.37 +1.47 | 74.39 +1.26 | 67.37 +1.54 | 89.38 +3.34 | 83.18 +6.06 |
| iVoro-NDAIL ★♠♣♦▼ | 77.57 +5.95 | 66.54 +5.40 | 72.50 +2.63 | 62.28 +1.91 | **81.24** +3.05 | **71.45** +1.10 | **81.74** +7.31 | **72.34** +5.00 | **80.22** +5.86 | **71.13** +3.76 | **79.08** +4.69 | **69.95** +2.58 | **90.04** +0.66 | **83.84** +0.66 |

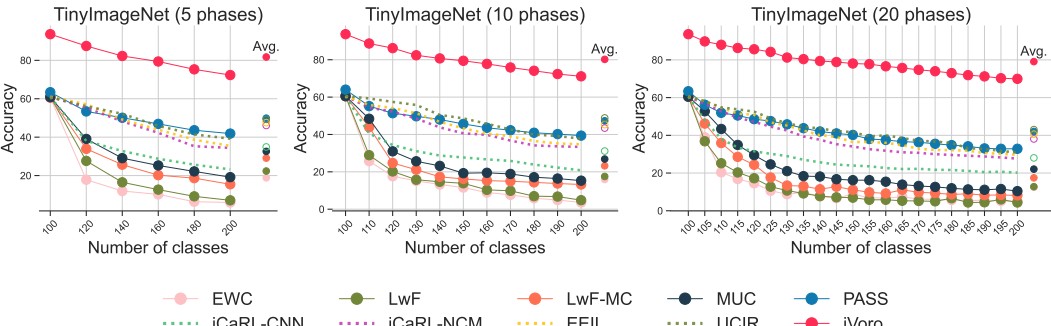

Figure G.1: Top-1 classification accuracy on TinyImagenet during 5/10/20 phases of CIL.

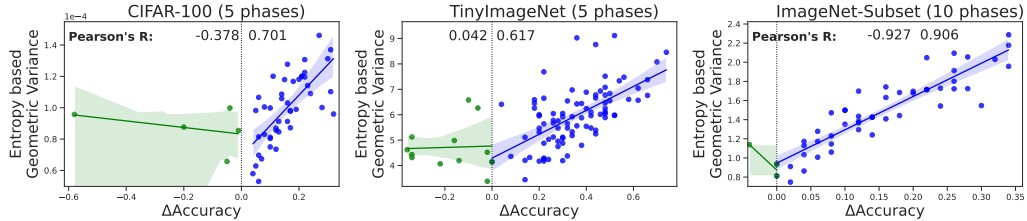

Figure G.2: Entropy-based geometric variance (HV) in class level as a function of $\Delta$accuracy.

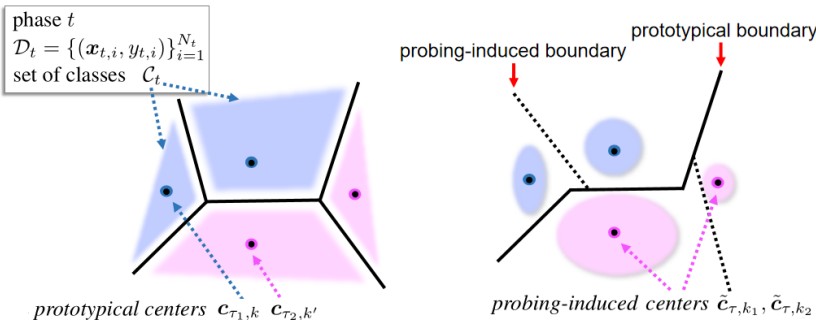

Figure G.3: Illustration of iVoro (left) and iVoro-D (right). iVoro treats all prototypes indifferently regardless of the phase, whereas iVoro-D refines the within-phase boundaries through a divide-and-conquer algorithm.

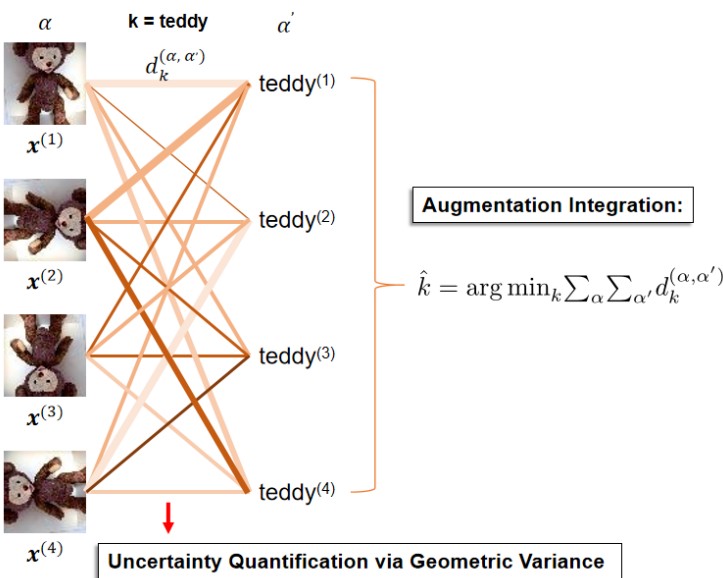

Figure G.4: Illustration of augmentation integration using a sample image from the "teddy" class. Each of the rotated images can possibly be assigned to each of the expanded labels (i.e. teddy$^{(1)}$, teddy$^{(2)}$, teddy$^{(3)}$, and teddy$^{(4)}$). The final score of "teddy" will be the aggregation of all the 16 predictions. The variance of the 16 predictions reflects the confidence of this prediction.

## H   DETAILED ANALYSIS ON PARAMETERIZED FEATURE NORMALIZATION

In this section we give a detailed comparison between iVoro (no normalization) and iVoro-N (parameterized feature normalization). On both CIFAR-100 and TinyImageNet with different phases, we show the distribution of accuracy (across all phases) for iVoro, iVoro-N with only $L_2$ normalization, and iVoro-N with both $L_2$ normalization and Tukey's ladder of powers transformation ($\lambda$ varying from 0.3 to 0.9). As shown in Fig. H.1, feature normalization benefits both datasets, but the efficacy is more prominent in more complex dataset e.g. TinyImageNet. Overall, iVoro-N improves the accuracy in the last phase by up to 1.65% on CIFAR-100, 2.21% on TinyImageNet, and 2.40 on ImageNet-Subset, respectively, compared to iVoro.

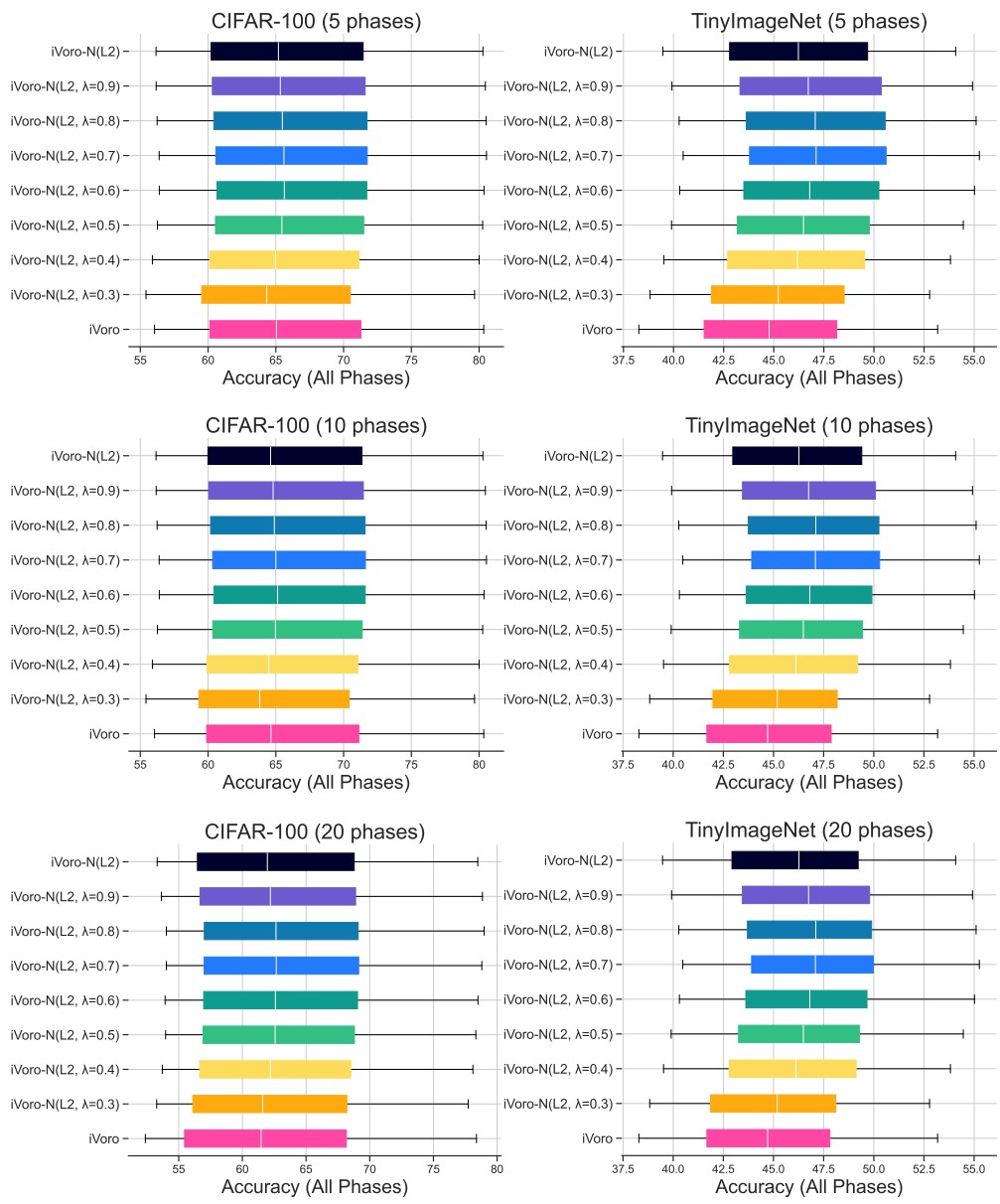

Figure H.1: Effect of parameterized feature transformation on CIFAR-100 and TinyImageNet.

# I UNCERTAINTY ANALYSIS

In this section, we demonstrate that the Entropy-based geometric variance (HV) is a good indicator of predictive uncertainty induced by SSL-based label augmentation, and also exhibits high correlation with predictive accuracy. Fig. I.1, Fig. I.2 and Fig. I.3 present the distribution of HV of each class during each phase (0 to 5) in 5-phase CIFAR-100 data. The color of the box reflects the accuracy of iVoro for the specific class. It can be observed in these figures that, the class with higher predictive accuracy will typically exhibit lower geometric uncertainty.

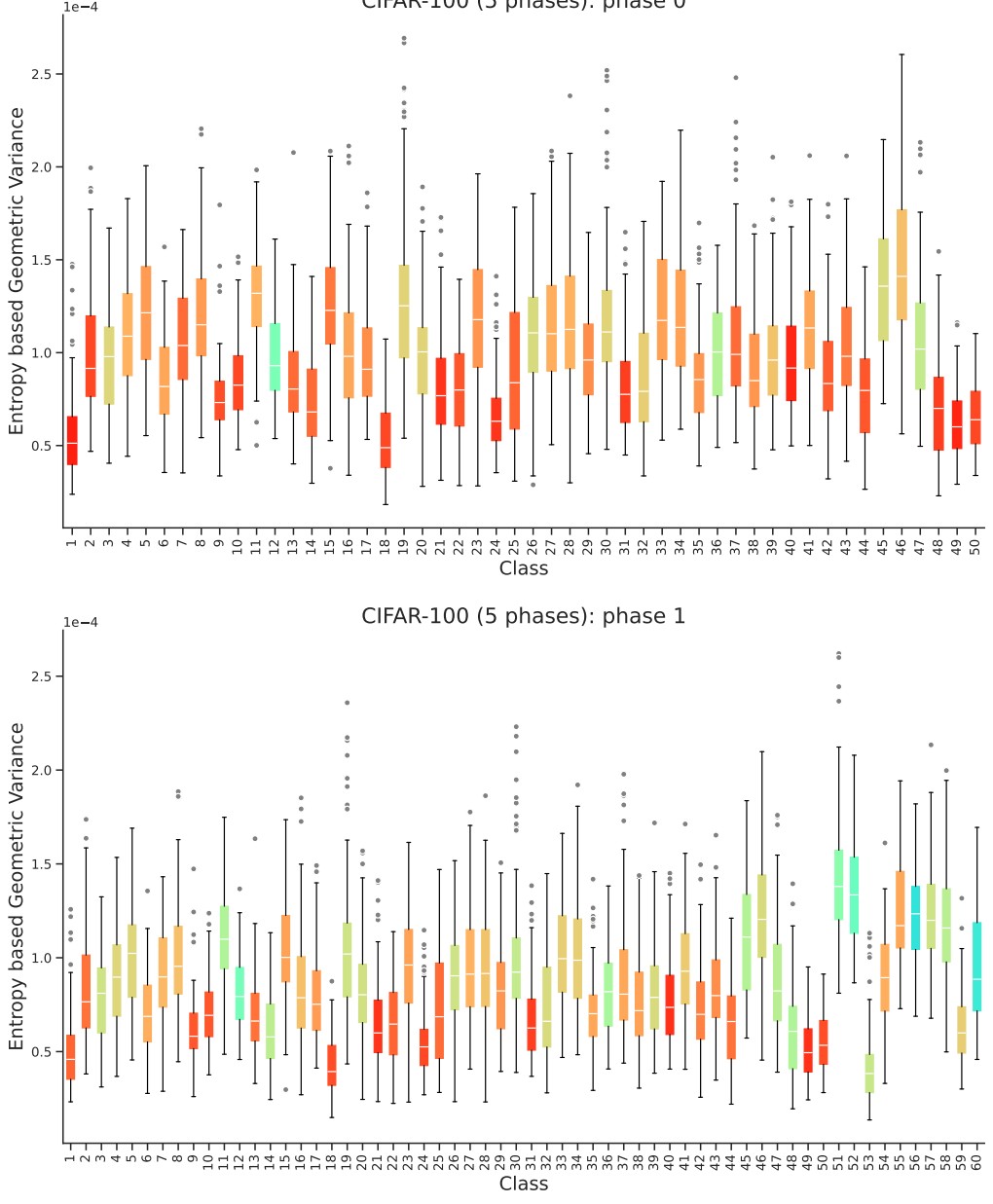

Figure I.1: The distributions of Entropy-based geometric variance for each class in each phase on 5-phase CIFAR-100 (phase 0 to phase 1).

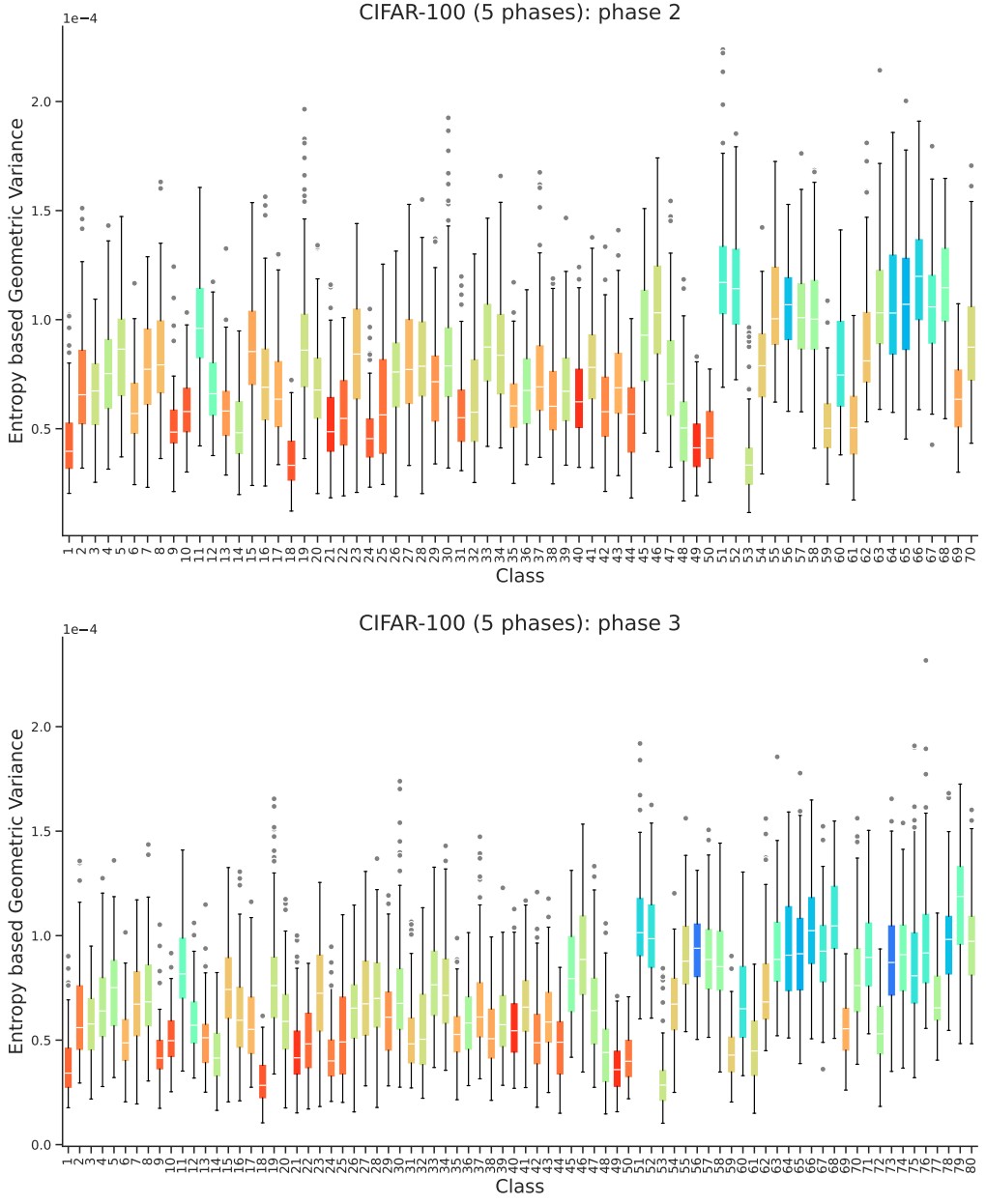

Figure I.2: The distributions of Entropy-based geometric variance for each class in each phase on 5-phase CIFAR-100 (phase 2 to phase 3).

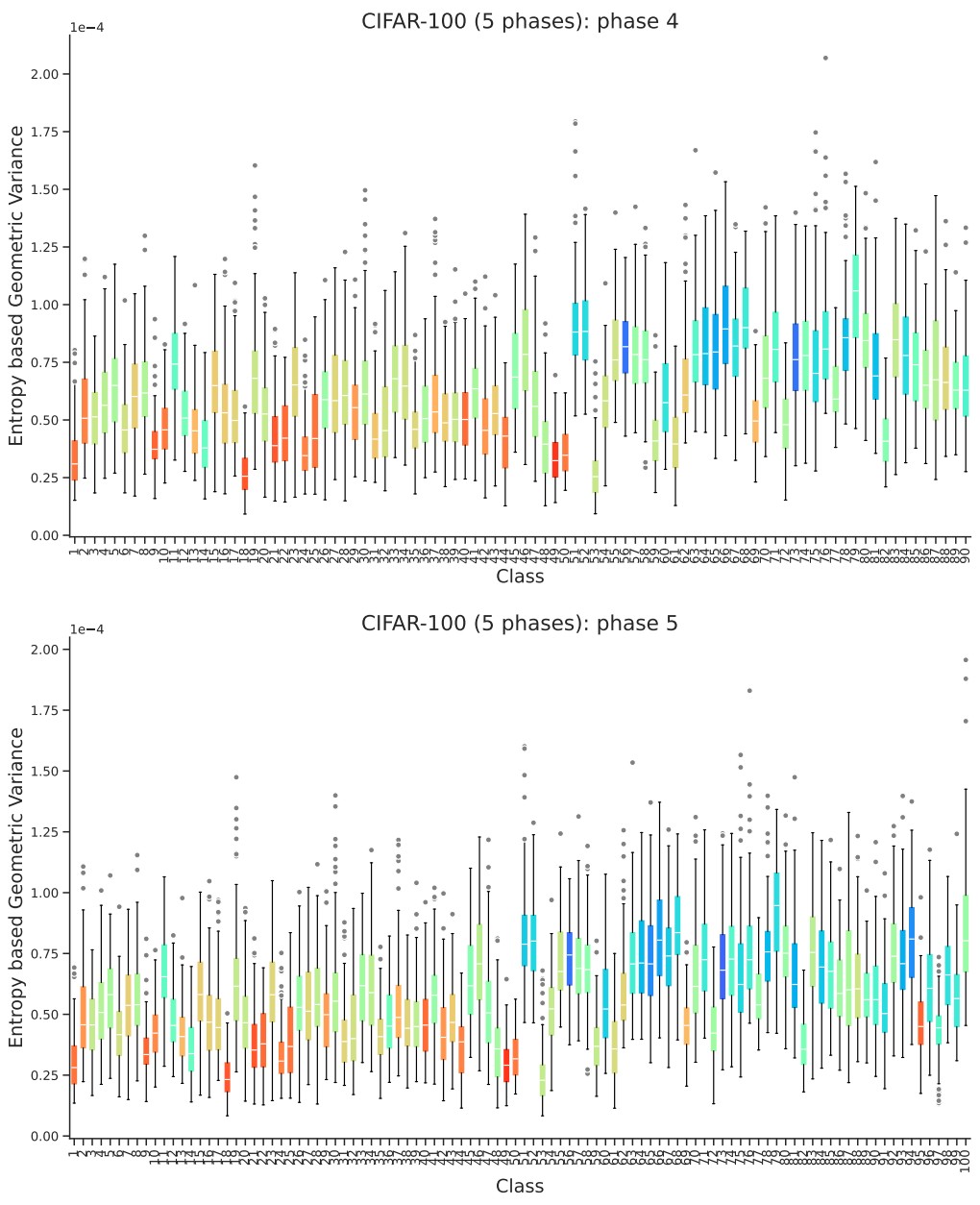

Figure I.3: The distributions of Entropy-based geometric variance for each class in each phase on 5-phase CIFAR-100 (phase 4 to phase 5).

## J    ANALYSIS ON THE FEATURE EXTRACTOR

In order to examine the effect of the feature extractor on the final result, we gradually decrease the number of classes in the first phase from 50 to 40, 30, 20, and 10 and still include 5 classes in each subsequent phase. When compared with PASS, the best version of iVoro still has 17.75%, 13.59%, 10.89%, and 1.60% improvements with 40, 30, 20, and 10 initial classes, respectively.

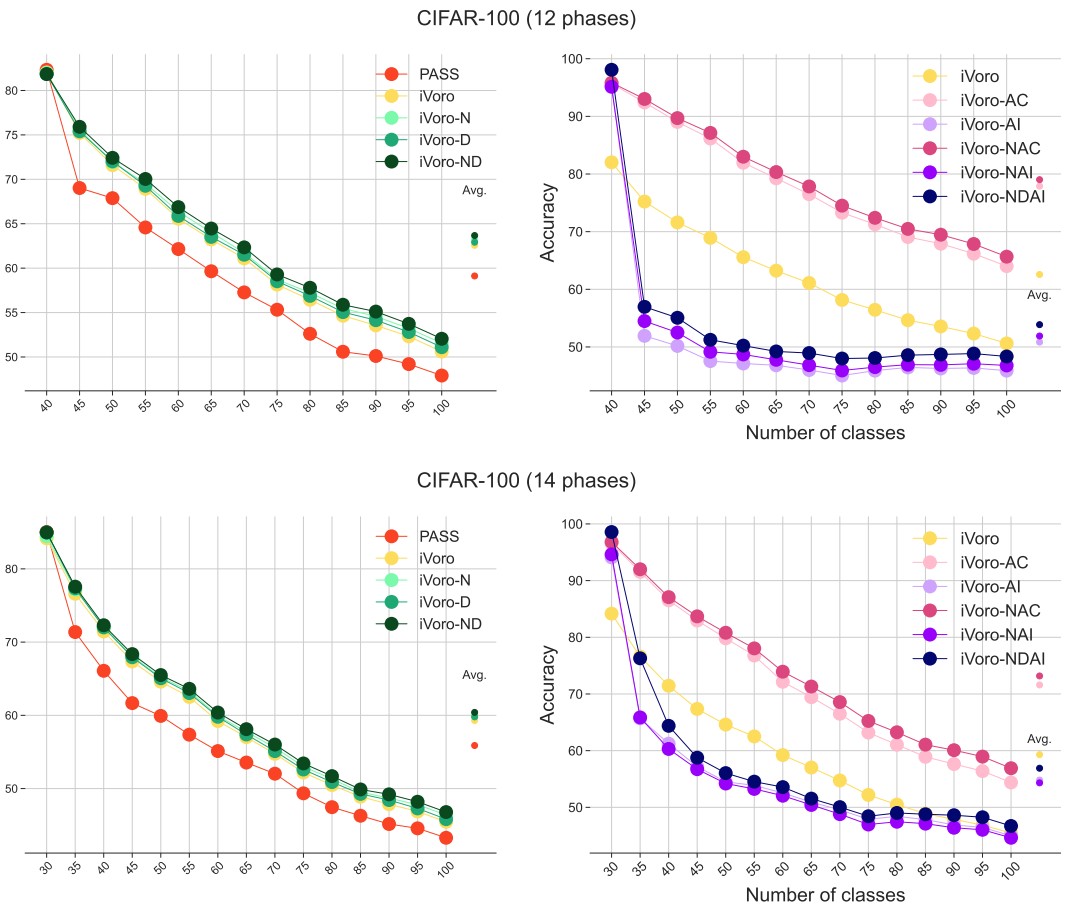

Figure J.1: Top-1 classification accuracy on CIFAR-100 during 12/14 phases of CIL, in which the number of classes in the first phase are decreased to 40/30, respectively.

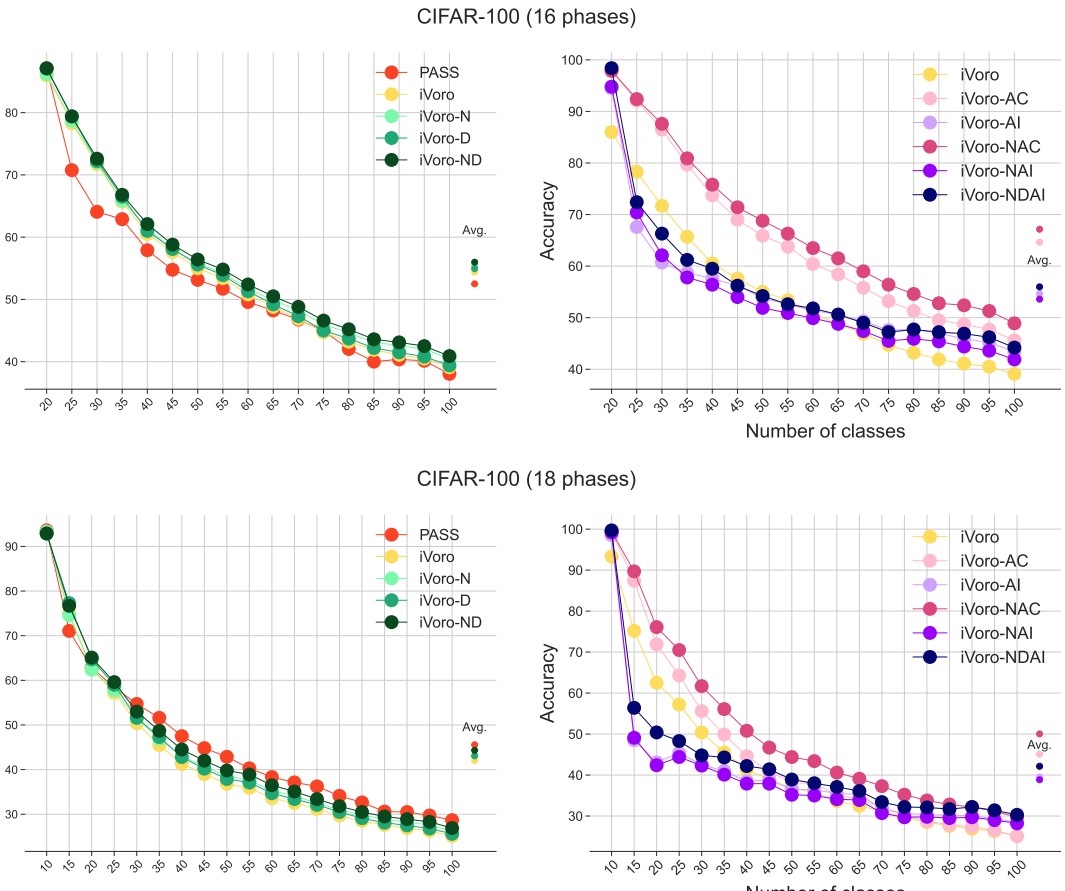

Figure J.2: Top-1 classification accuracy on CIFAR-100 during 16/18 phases of CIL, in which the number of classes in the first phase are decreased to 20/10, respectively.

## K ANALYSIS ON THE MULTILAYER VORONOI DIAGRAMS

In this section, we extract the feature from the $3^{rd}$ block and rebuild iVoro, to be iVoro3, etc. The results for iVoro3, iVoro3-N, iVoro3-NAC, and iVoro3-NAI are shown in Fig. F.2, Fig. F.3, and Fig. F.1. As expected, there is always a substantial decrease in accuracy if only the features from the $3^{rd}$ block are used. For example, iVoro drops from 38.27% to 18.71%, from 56.05% to 43.22%, and from 55.40% to 35.10% on TinyImageNet, CIFAR-100, and ImageNet-Subset, respectively. However, when augmentation integration is used, the accuracy of iVoro3-NAI becomes 42.03% (TinyImageNet), 66.13% (CIFAR-100), and 64.32% (ImageNet-Subset), even much higher than iVoro and PASS. Moreover, when integrating the features from $\phi$ and $\phi^{(3)}$ using CCVD, iVoro-NDAIL achieves the best performance across all variants of iVoro, 72.34% on TinyImageNet and 83.84% on ImageNet-Subset.

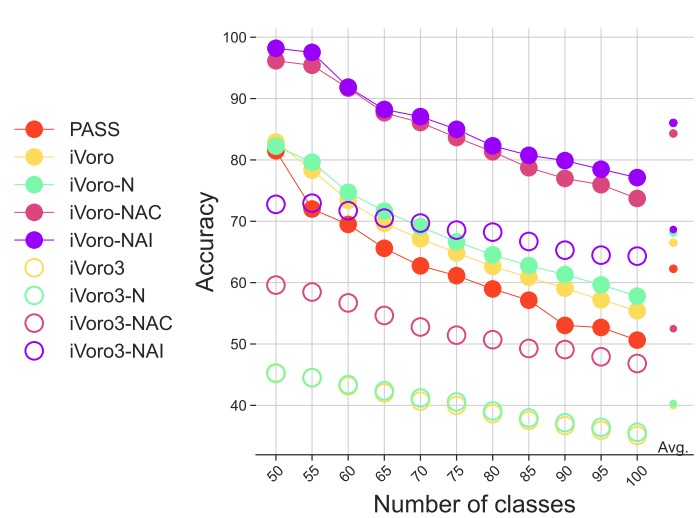

Figure K.1: Comparison between iVoros constructed from the final block (iVoro) and the $3^{rd}$ block (iVoro3) w.r.t. the top-1 classification accuracy on ImageNet-Subset during 10 phases of CIL.

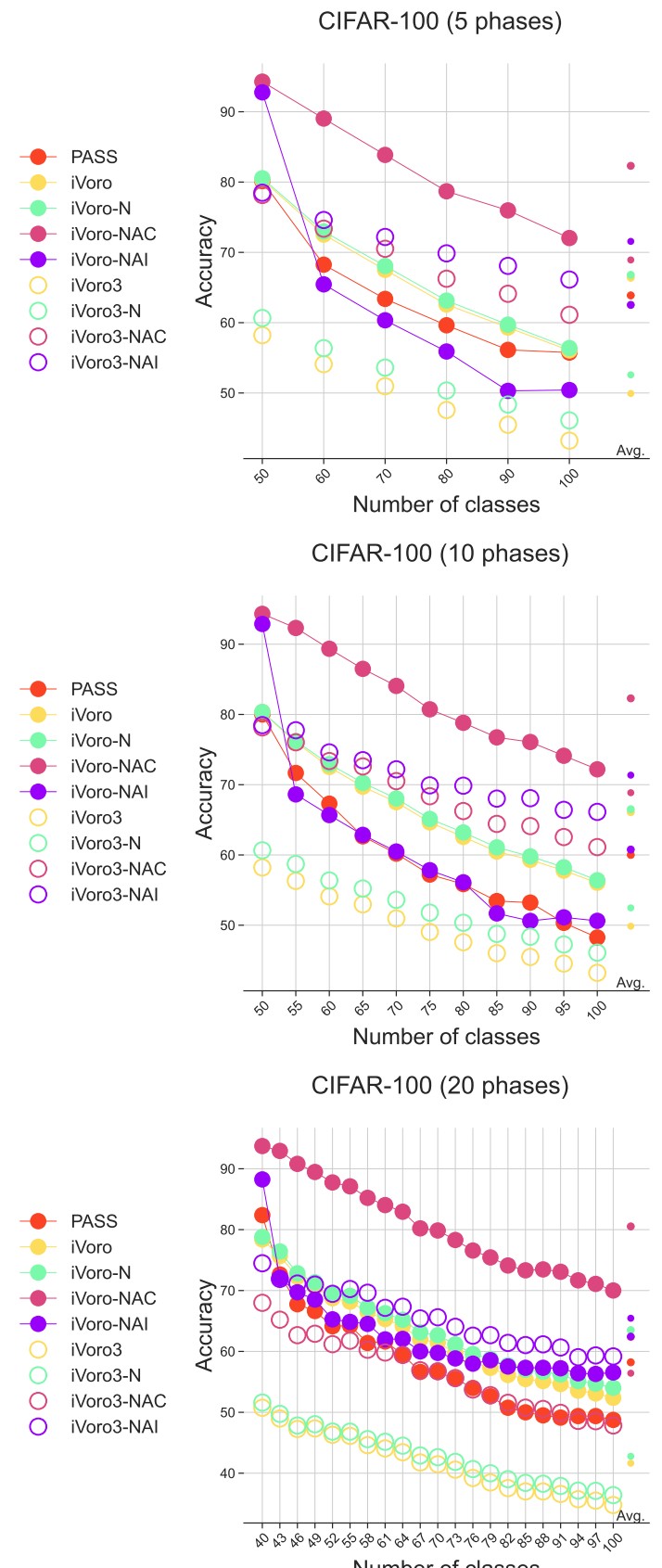

Figure K.2: Comparison between iVoros constructed from the final block (iVoro) and the $3^{rd}$ block (iVoro3) w.r.t. the top-1 classification accuracy on CIFAR100 during 5/10/20 phases of CIL.

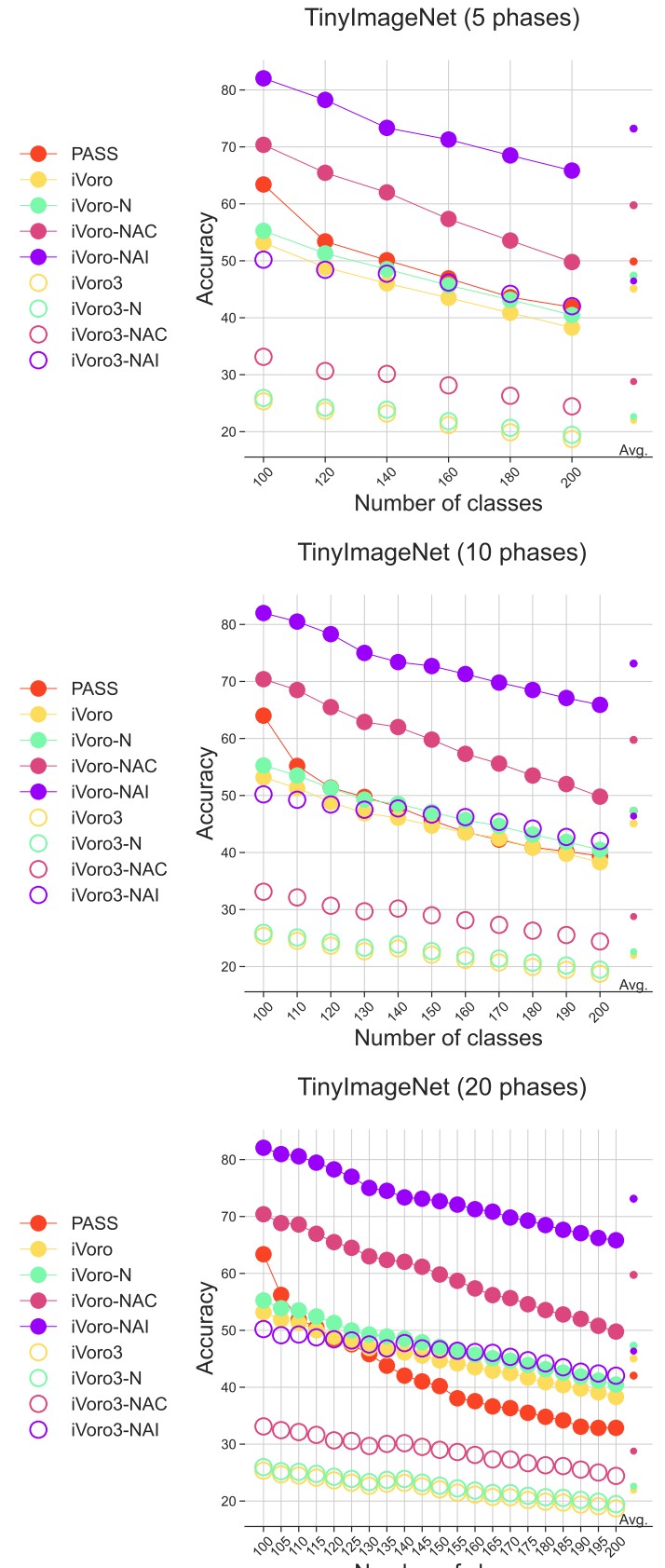

Figure K.3: Comparison between iVoros constructed from the final block (iVoro) and the $3^{rd}$ block (iVoro3) w.r.t. the top-1 classification accuracy on TinyImagenet during 5/10/20 phases of CIL.

## L    COMPLETE LIST OF ALGORITHMS

Here we provide the algorithm for Voronoi Diagram reduction (Ma et al., 2022a) in Alg. 1, the iVoro algorithm in Alg. 2, the iVoro-D algorithm in Alg. 3.

---

**Algorithm 1:** Voronoi Diagram-based Logistic Regression.

**Data:** Local data $\mathcal{D}_t$ in phase $\tau$
**Result:** $\boldsymbol{W}_\tau$
1  Initialize $\boldsymbol{W}_\tau \leftarrow \boldsymbol{W}_\tau^{(0)}$;
2  **for** *epoch $\leftarrow 1, ..., $#epoch* **do**
3      $b_{\tau,k} \leftarrow -\frac{1}{4}||\boldsymbol{W}_{\tau,k}||_2^2, \forall k = 1, ..., K_\tau$ ;                    ◁ Apply Theorem 2.1
4      Compute loss $\mathcal{L}(\boldsymbol{W}_\tau, \boldsymbol{b}_\tau)$ ;                    ◁ forward propagation
5      Update $\boldsymbol{W}_\tau$ ;                    ◁ backward propagation
6  **end**
7  **return** $\boldsymbol{W}_\tau$

---

**Algorithm 2:** iVoro Algorithm.

**Data:** Training datasets until phase $t$:
        $\mathcal{D}_\tau = \{(\boldsymbol{x}_{\tau,i}, y_{\tau,i})\}_{i=1}^{N_\tau}, \boldsymbol{x}_{\tau,i} \in \mathbb{D}, y_{\tau,i} \in \mathcal{C}_\tau, \tau \in \{1, ..., t\}$, query example $\boldsymbol{x}$
**Result:** prediction $\hat{y}$
1  **for** $\tau \in \{1, ..., t\}$ **do**
2      **for** $k \in \{1, ..., K_\tau\}$ **do**
3          $\boldsymbol{c}_{\tau,k} \leftarrow \frac{1}{N_{\tau,k}} \sum_{i\in\{1,...,N_{\tau,k}\},y=k} \phi(\boldsymbol{x}_{\tau,i})$ ;                    ◁ prototypical centers
4          $\nu_{\tau,k} \leftarrow 0$
5      **end**
6  **end**
7  $\boldsymbol{z} \leftarrow \phi(\boldsymbol{x})$
8  $\boldsymbol{z} \leftarrow (h_\lambda \circ g_{w,\eta} \circ f)(\boldsymbol{z})$ ;                    ◁ iVoro-N, optional
9  $\hat{y} \leftarrow \mathcal{C}_{\tau',k'}$ s.t. $d(\boldsymbol{z}, \boldsymbol{c}_{\tau',k'}) = \min_{\tau,k} d(\boldsymbol{z}, \boldsymbol{c}_{\tau,k})$
10 **return** $\{\boldsymbol{c}_{\tau,k}\}, \hat{y}$

---

---

**Algorithm 3:** iVoro-D Algorithm. The time complexity for the establishment of VD is $\mathcal{O}((\sum_{\tau=1}^{t} K_\tau)^2)$, the time complexity for querying the VD is $\mathcal{O}(\sum_{\tau=1}^{t} K_\tau)$.

---

**Data:** Training datasets until phase $t$:
$\mathcal{D}_\tau = \{(\boldsymbol{x}_{\tau,i}, y_{\tau,i})\}_{i=1}^{N_\tau}, \boldsymbol{x}_{\tau,i} \in \mathbb{D}, y_{\tau,i} \in \mathcal{C}_\tau, \tau \in \{1, ..., t\}$, query example $\boldsymbol{x}$
**Result:** Prediction $\hat{y}$

1 **for** $\tau \in \{1, ..., t\}$ **do**
2      $\boldsymbol{W}_\tau, \boldsymbol{b}_\tau \leftarrow$ Algorithm 1($\mathcal{D}_\tau$)
3      $\tilde{\boldsymbol{c}}_{\tau,k} \leftarrow \frac{1}{2}\boldsymbol{W}_{\tau,k}$ ;                                    ◁ probing-induced centers
4      $\boldsymbol{c}_{\tau,k} \leftarrow$ Algorithm 2($\mathcal{D}_\tau$) ;                                 ◁ prototypical centers
5      **for** $\mathcal{C}_{\tau,k_1}, \mathcal{C}_{\tau,k_2} \in \mathcal{C}_\tau$ **do**
6          $\boldsymbol{v} \leftarrow \frac{\tilde{\boldsymbol{c}}_{\tau,k_1} - \tilde{\boldsymbol{c}}_{\tau,k_2}}{||\tilde{\boldsymbol{c}}_{\tau,k_1} - \tilde{\boldsymbol{c}}_{\tau,k_2}||_2}$
7          $q \leftarrow \frac{||\tilde{\boldsymbol{c}}_{\tau,k_1}||_2^2 - ||\tilde{\boldsymbol{c}}_{\tau,k_2}||_2^2}{2||\tilde{\boldsymbol{c}}_{\tau,k_1} - \tilde{\boldsymbol{c}}_{\tau,k_2}||_2}$ ;                  ◁ within-clique boundaries
8      **end**
9 **end**
10 **for** $\mathcal{C}_{\tau_1,k} \in \mathcal{C}_{\tau_1}, \mathcal{C}_{\tau_2,k'} \in \mathcal{C}_{\tau_2}$ **do**
11      $\boldsymbol{v} \leftarrow \frac{\boldsymbol{c}_{\tau_1,k} - \boldsymbol{c}_{\tau_2,k'}}{||\boldsymbol{c}_{\tau_1,k} - \boldsymbol{c}_{\tau_2,k'}||_2}$
12      $q \leftarrow \frac{||\boldsymbol{c}_{\tau_1,k}||_2^2 - ||\boldsymbol{c}_{\tau_2,k'}||_2^2}{2||\boldsymbol{c}_{\tau_1,k} - \boldsymbol{c}_{\tau_2,k'}||_2}$ ;                  ◁ cross-clique boundaries
13 **end**
14 **for** $\Gamma \in \{\Gamma\}$ **do**
15      Delete candidate and its boundaries w.r.t. $\text{sign}(\boldsymbol{v}^T\boldsymbol{z} - q)$
16      **if** *only one candidate* $\mathcal{C}_{\tau',k'}$ *remains* **then**
17          $\hat{y} \leftarrow \mathcal{C}_{\tau',k'}$
18      **end**
19 **end**
20 **return** $\hat{y}$

---

# M    FORGETTING ANALYSIS

Following PASS (Zhu et al., 2021), we also quantitatively measure the degree of the catastrophic forgetting. Specifically, at phase $t$, the accuracy drop from the maximum accuracy to current accuracy for datasets from each phase $\tau \in \{1, ..., t\}$ is denoted as the average forgetting, as shown in Fig. M.1 and Fig. M.2 for CIFAR-100 and TinyImageNet, respectively. For PASS, the average forgetting keeps growing during phases. However, for iVoro/iVoro-N/iVoro-NAC/iVoro-NAI, the catastrophic forgetting is significantly overcome. For example, on CIFAR-100 (5 phase), forgetting in the last phase is decreased from 20.28 to 8.17/8.95/5.92 for iVoro/iVoro-N/iVoro-NAC. For a local dataset $\mathcal{D}_\tau = \{(\boldsymbol{x}_{\tau,i}, y_{\tau,i})\}_{i=1}^{N_\tau}, \boldsymbol{x}_{\tau,i} \in \mathbb{D}, y_{\tau,i} \in \mathcal{C}_\tau$ and a set of fixed prototypes $\{\boldsymbol{c}_{\tau,k}\}_{\tau \in \{1,...,t\}, k \in \{1,...,K_\tau\}}$, the prediction $\hat{y} = \arg\min_{k \in \{1,...,K_\tau\}} ||\phi(\boldsymbol{x}) - \boldsymbol{c}_{\tau,k}||_2^2$ is less likely to change compared to continuously updated model, as in PASS, implying that, aligning with our intuition (Sec. 1), the VD structure can naturally and successfully combat catastrophic forgetting.

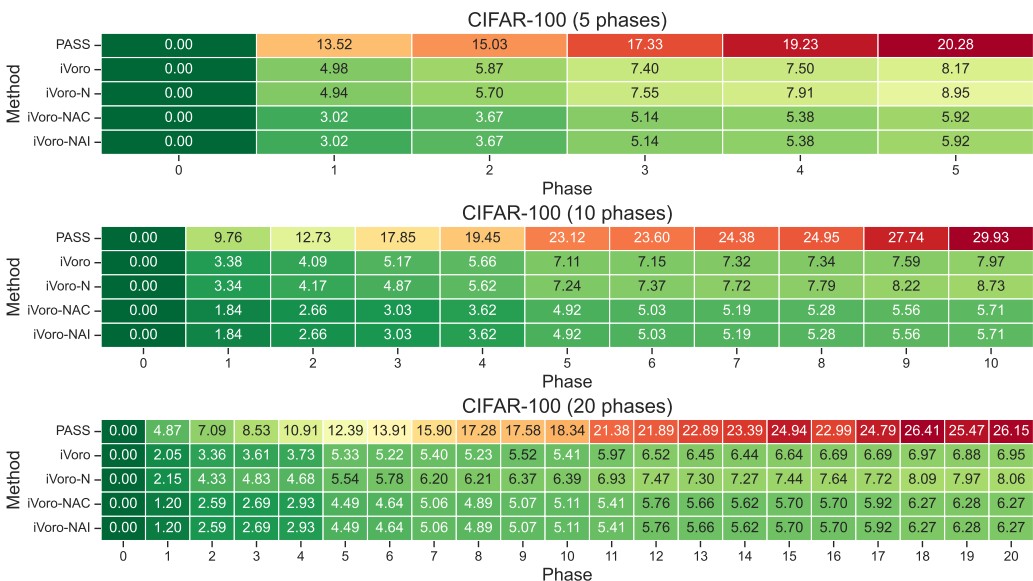

Figure M.1: Results of average forgetting on CIFAR-100.

Table M.1: Average Forgetting ($\downarrow$): Comparison between iVoro with state-of-the-art CIL methods

| | **CIFAR100** | | | **TinyImageNet** | | |
|---|---|---|---|---|---|---|
| Methods | 5 phases | 10 phases | 20 phases | 5 phases | 10 phases | 20 phases |
| ✔ iCaRL$_{\text{CNN}}$ (Rebuffi et al., 2017) | 42.13 | 45.69 | 43.54 | 36.89 | 36.7 | 45.12 |
| ✔ iCaRL$_{\text{CNN}}$ (Rebuffi et al., 2017) | 24.90 | 28.32 | 35.53 | 27.15 | 28.89 | 37.40 |
| ✔ EEIL (Castro et al., 2018) | 23.36 | 26.65 | 32.40 | 25.56 | 25.91 | 35.04 |
| ✔ UCIR (Hou et al., 2019) | 21.00 | 25.12 | 28.65 | 20.61 | 22.25 | 33.74 |
| ✘ LwF-MC (Li & Hoiem, 2017) | 44.23 | 50.47 | 55.46 | 54.26 | 54.37 | 63.54 |
| ✘ MUC (Liu et al., 2020b) | 40.28 | 47.56 | 52.65 | 51.46 | 50.21 | 58.00 |
| ✘ PASS (Zhu et al., 2021) | 20.28 | 29.93 | 26.15 | 8.91 | 23.11 | 30.55 |
| ✘ iVoro (ours) | **8.17** | **7.97** | **6.95** | **5.68** | **5.55** | **5.34** |

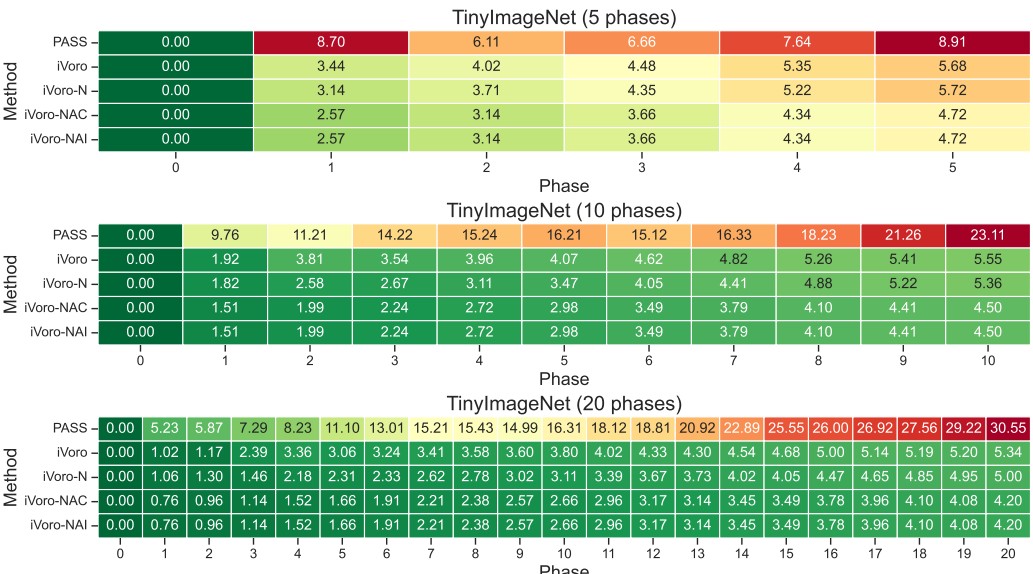

Figure M.2: Results of average forgetting on TinyImageNet.

# N    CLASS-LEVEL ANALYSIS

In order to examine the reason for our strong two-digit improvement (25.26%, 37.09%, and 33.21% for CIFAR-100, TinyImageNet, and ImageNet-Subset, respectively), in this section (Fig. N.1, Fig. N.2, Fig. N.3, Fig. N.4, Fig. N.5, Fig. N.6, Fig. N.7, and Fig. N.8), we illustrate the class-level accuracies along the 5 phases for all the 200 classes in TinyImageNet. Generally, iVoro is competitive to PASS, and iVoro-NAC/NAI performance significantly better than the former two. This is more evident for novel classes (i.e. class 100-199), showing that augmentation consensus/integration is particularly useful for unseen classes.

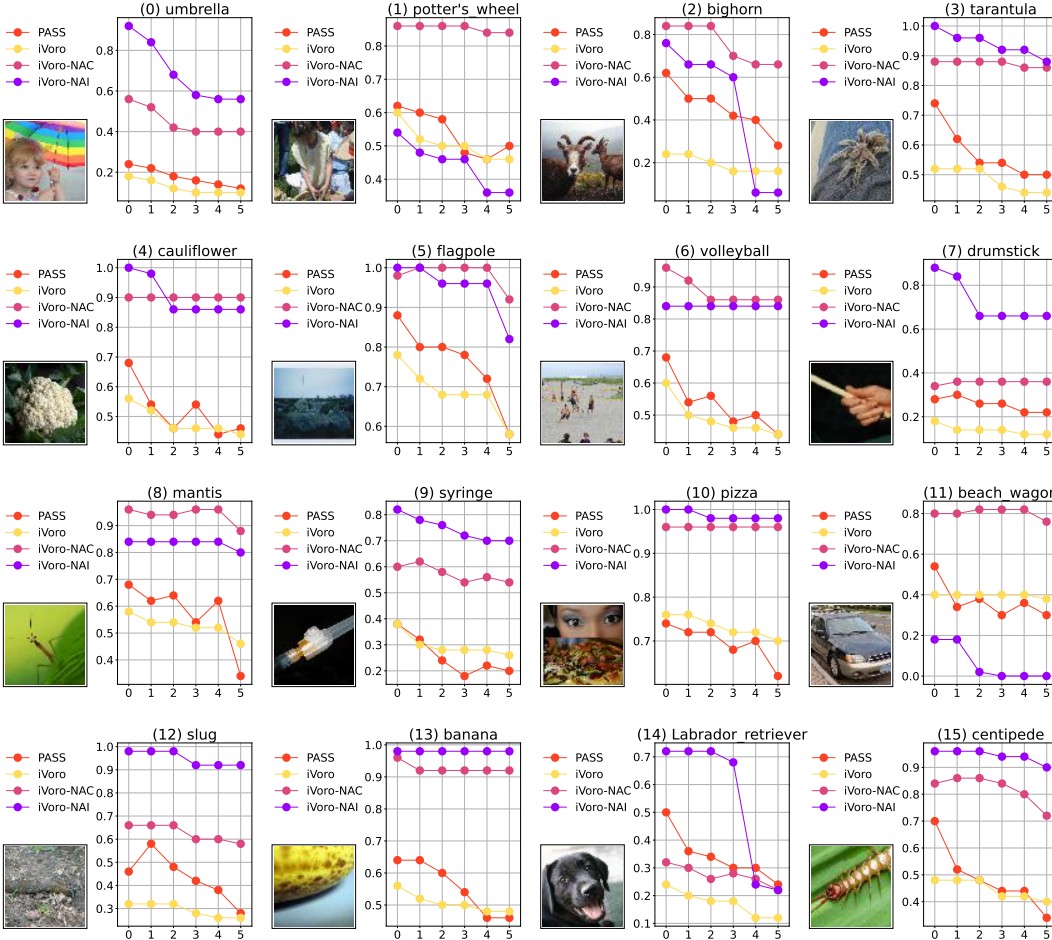

Figure N.1: Class-level comparison of PASS, iVoro, iVoro-NAC, and iVoro-NAI on the TinyImageNet dataset (5 phases). The x axis and y axis denote the phase index and the average accuracy in this class, respectively. A arbitrarily selected sample image class is also shown for every class.

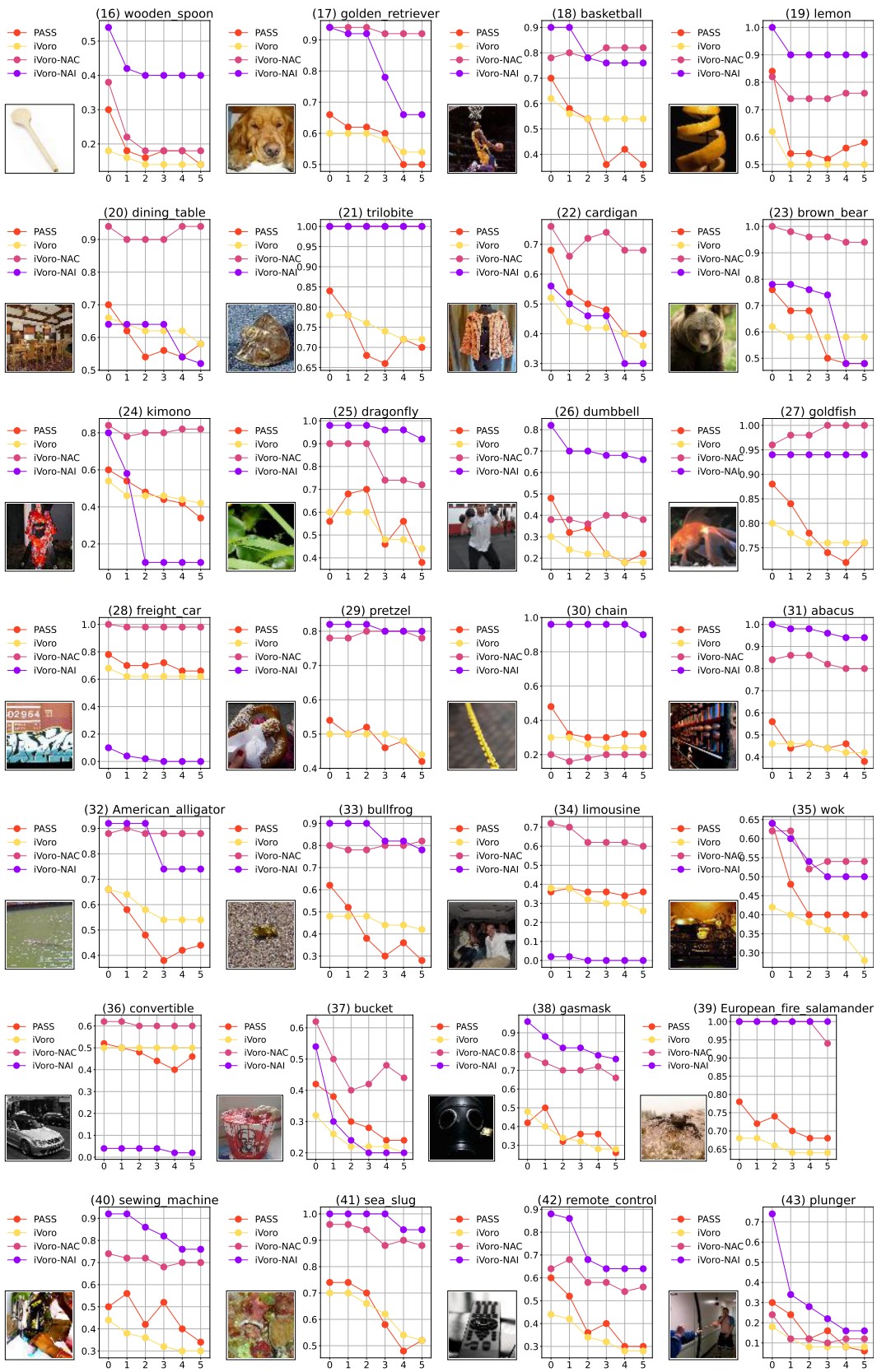

Figure N.2: Fig. N.1 continued.

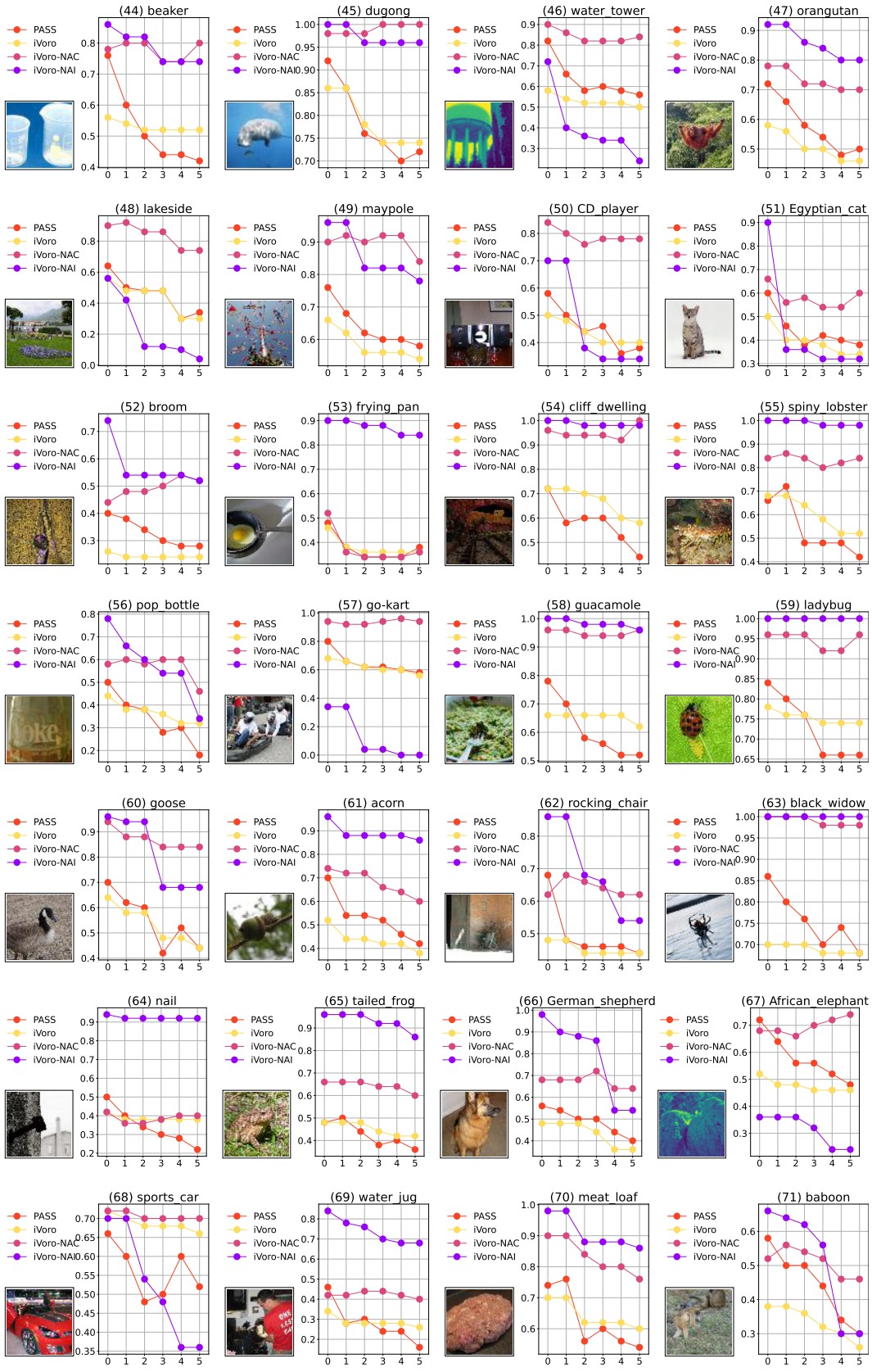

Figure N.3: Fig. N.1 continued.

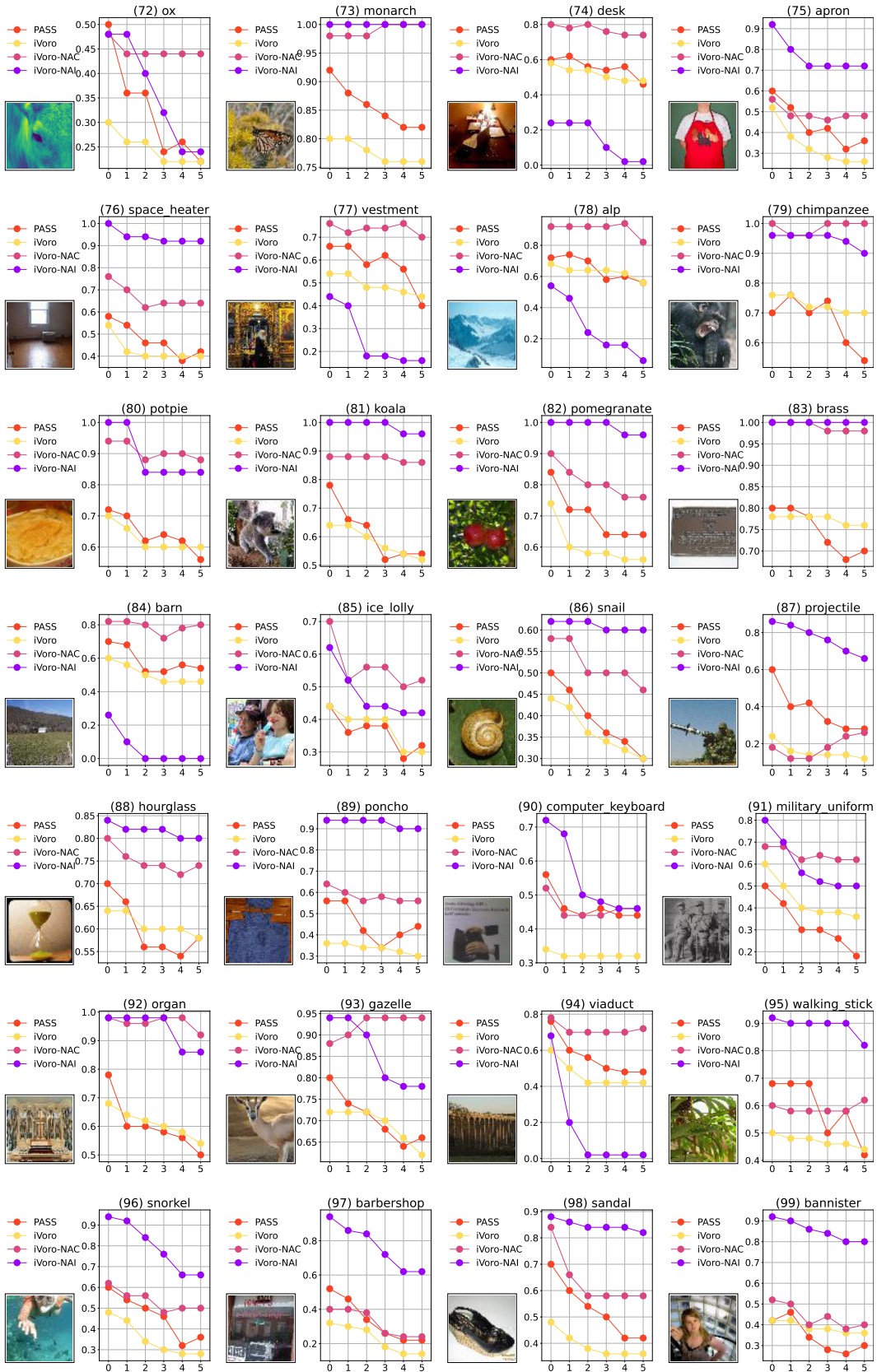

Figure N.4: Fig. N.1 continued.

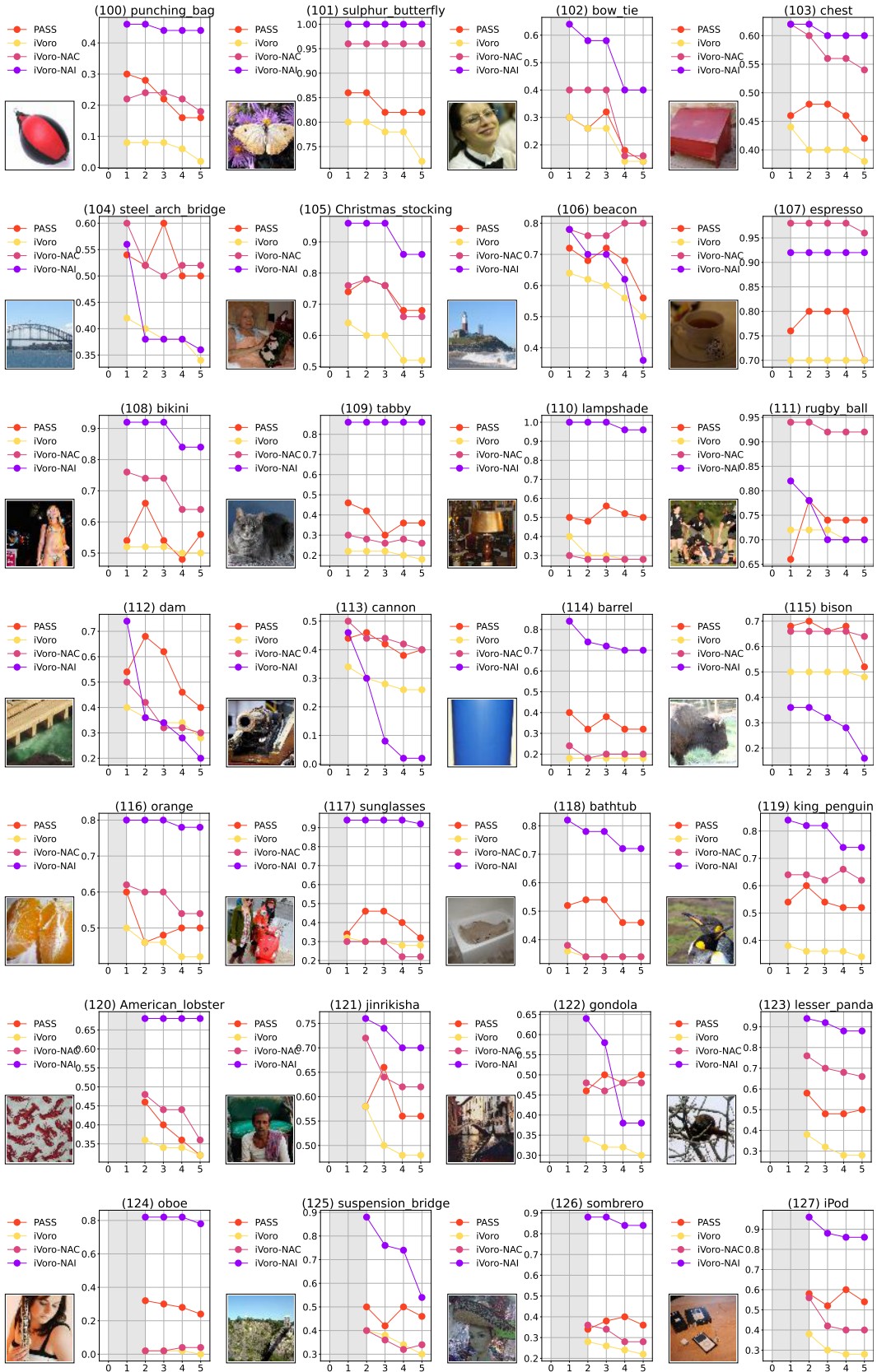

Figure N.5: Fig. N.1 continued.

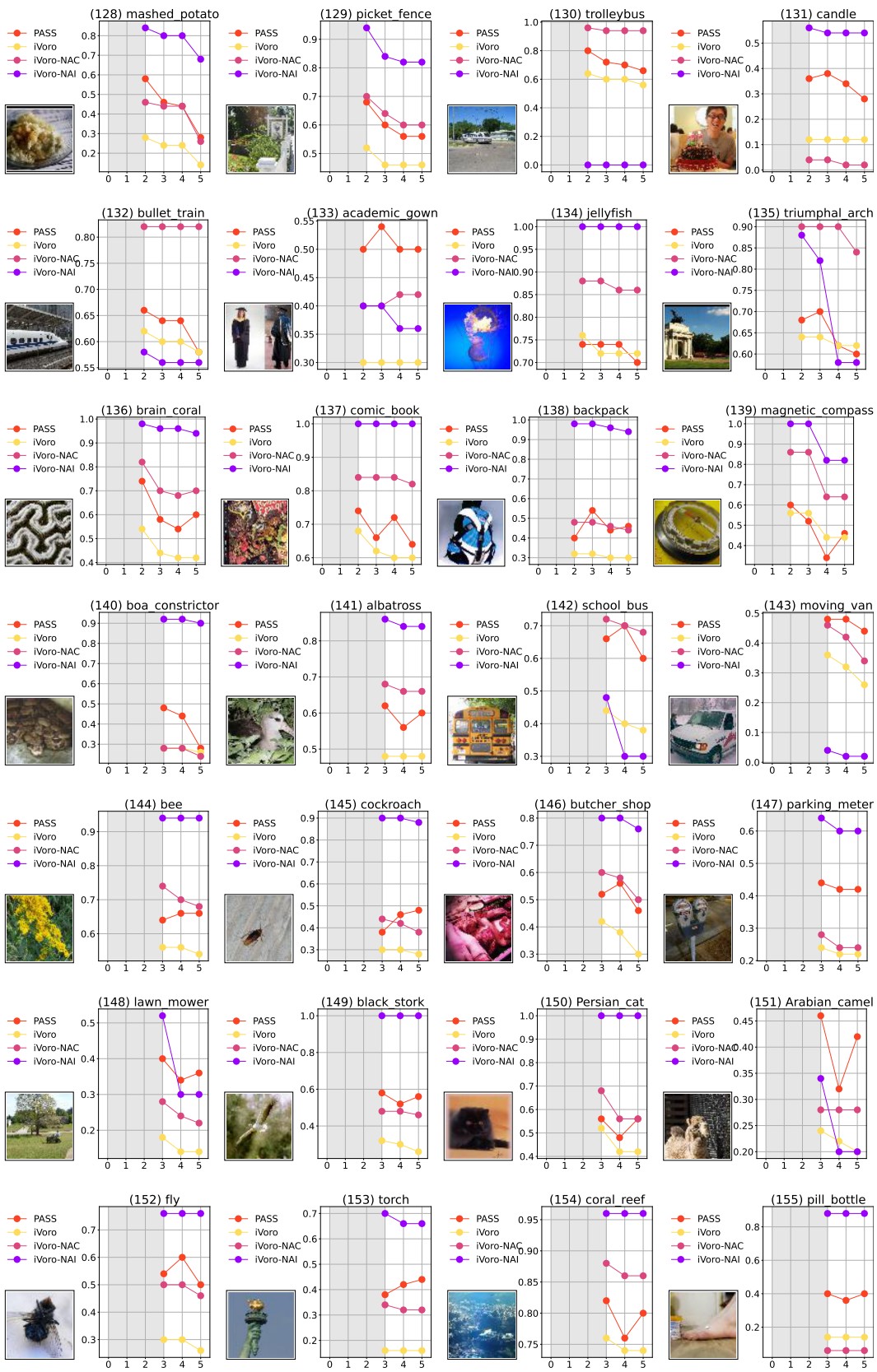

Figure N.6: Fig. N.1 continued.

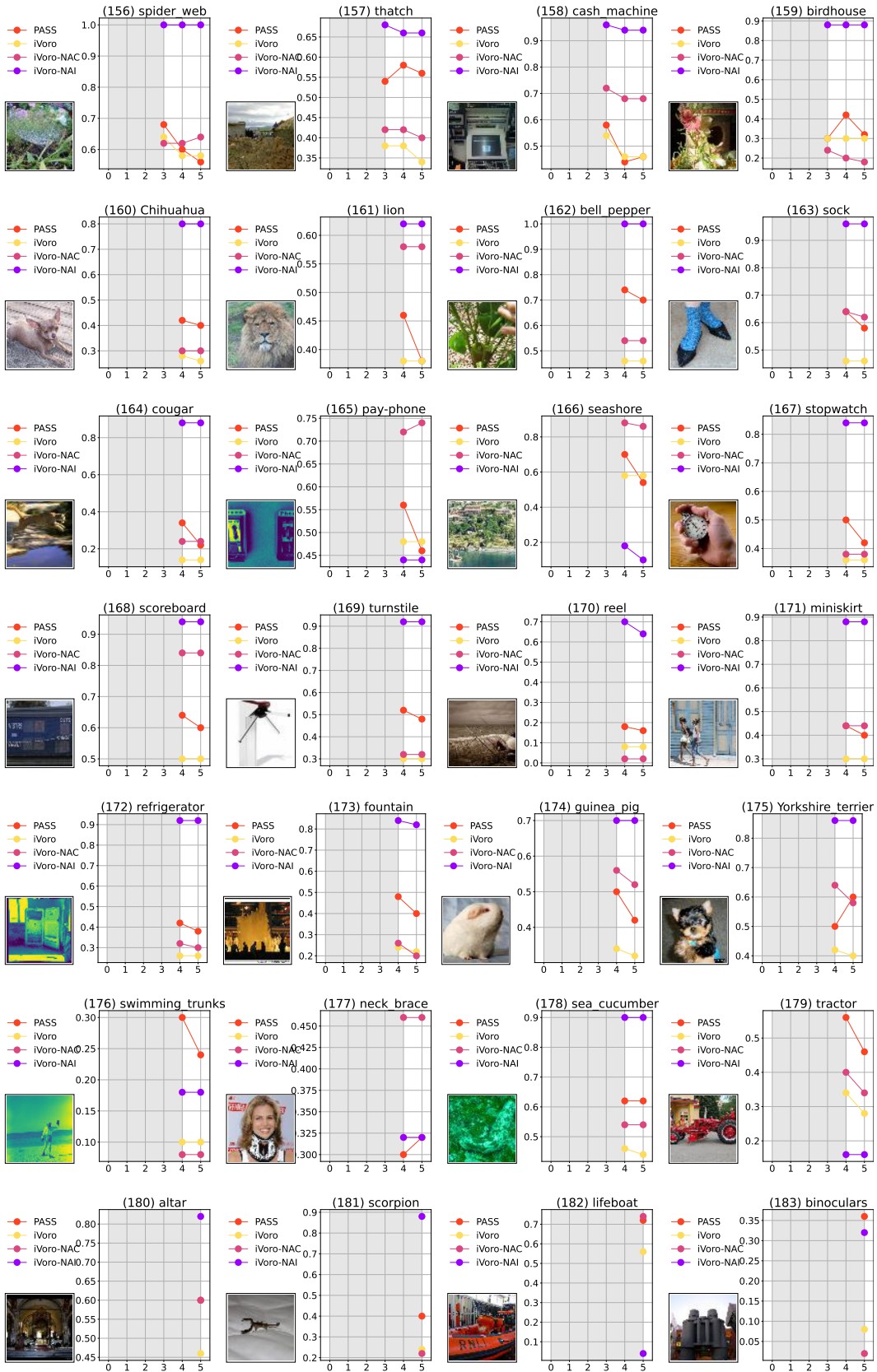

Figure N.7: Fig. N.1 continued.

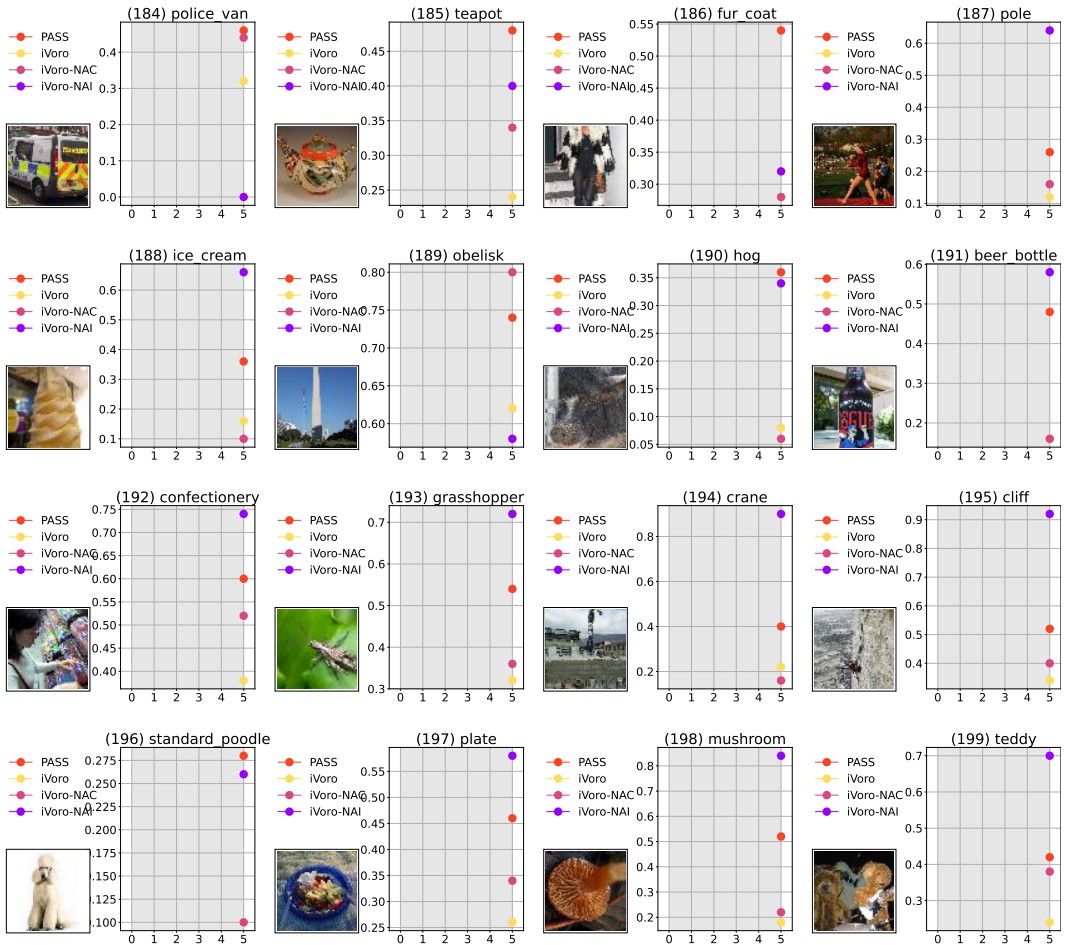

Figure N.8: Fig. N.1 continued.

## O   SAMPLE-LEVEL ANALYSIS

In Sec. N, we have analyzed the class-level improvement, and demonstrated that most of the time iVoro-NAI can surpass others by virtue of the postprocessing of the predictions from SSL-based label augmentation. In this section, we go into more detail at the sample level aiming at revealing why a misclassification can be corrected once our method is applied.

To do so, we *arbitrarily* inspect two examples from the first class (i.e. "umbrella"), and show them in Fig. O.1 and Fig. O.2, respectively.

In Fig. O.1, "umbrella" is within the top-3 labels predicted by PASS, which is, however, overwhelmed by "ice_lolly". From row 2 to row 9, label set I/II/III/IV represent the expended labels (I for the original, and II/III/IV for the 90/180/270-degree rotated, see Fig. G.4), and the distances from the query image (original or rotated) to the 100 classes (original or expanded) are shown. Interestingly, the original image is predicted as the same label ("ice_lolly") by iVoro. However, for the rotated images, the mis-predicted labels are various, "abacus", "bannister", "limousine", "vestment", "sewing_machine", "plunger", and "dumbbell". But importantly, the distance to the ground-truth label "umbrella" is always *relatively* small. Hence, when integrated altogether, the ground-truth label "umbrella" achieves the nearest distance (see upper right figure).

Similarly, in Fig. O.2, the sample is misclassified as "beach_wagon" by PASS, and is still misclassified by iVoro as "volleyball". However, when augmentation integration (iVoro-AI) is applied, the accumulated distances clearly direct to the correct label "umbrella".

In conclusion, these sample-level analyses show that iVoro-AC/AI elevates the performance by substantially enhancing the robustness of the (augmented) prediction – the rotated image set and the expanded label set collectively contribute to the strong two-digit improvement.

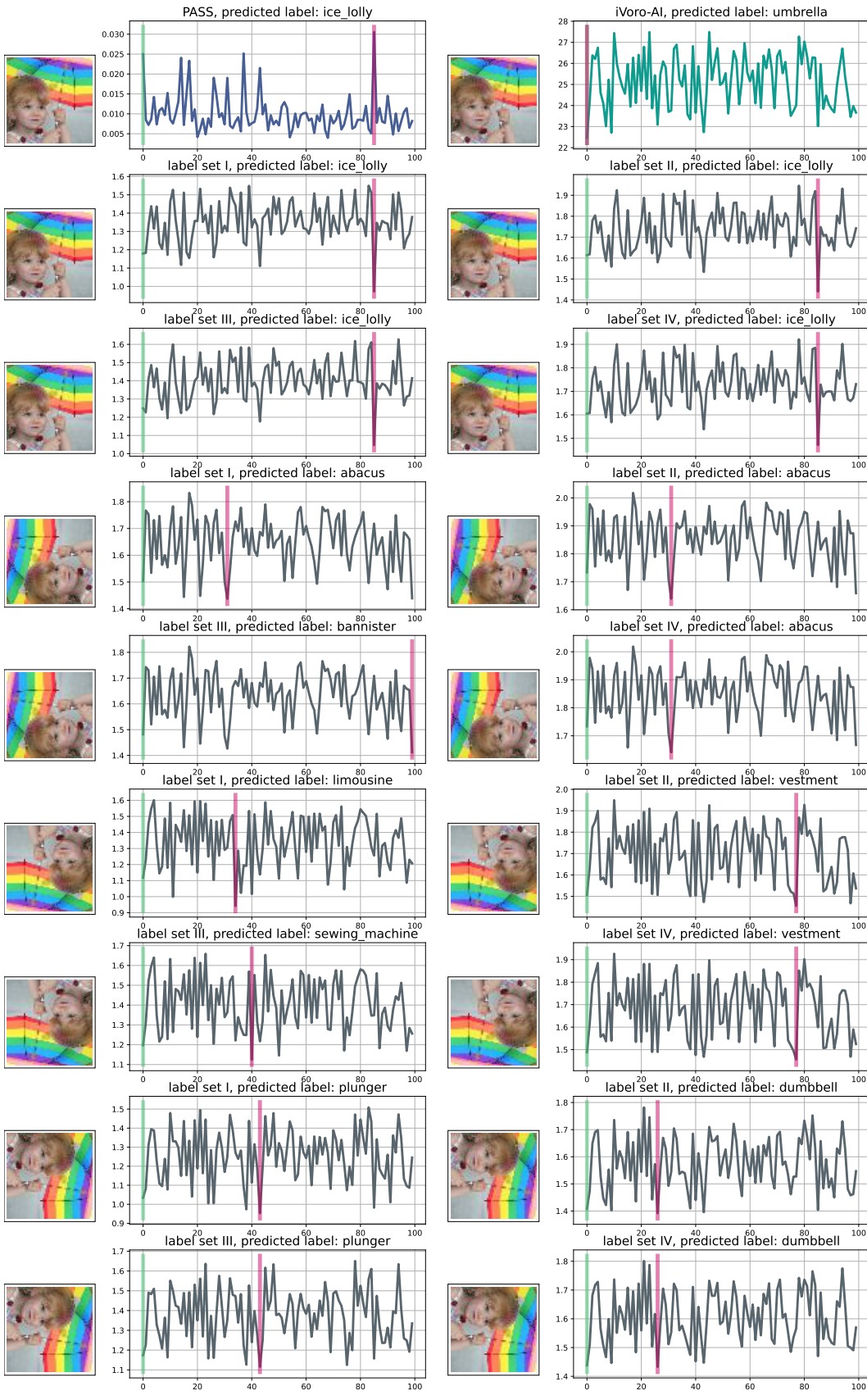

Figure O.1: Sample-level analysis of one example from the "umbrella" class. The y axis denotes the logit for 100 classes for PASS, and denotes the distance to 100 prototypes for iVoro. The green line indicates the ground-truth label while the red line denotes the predicted label.

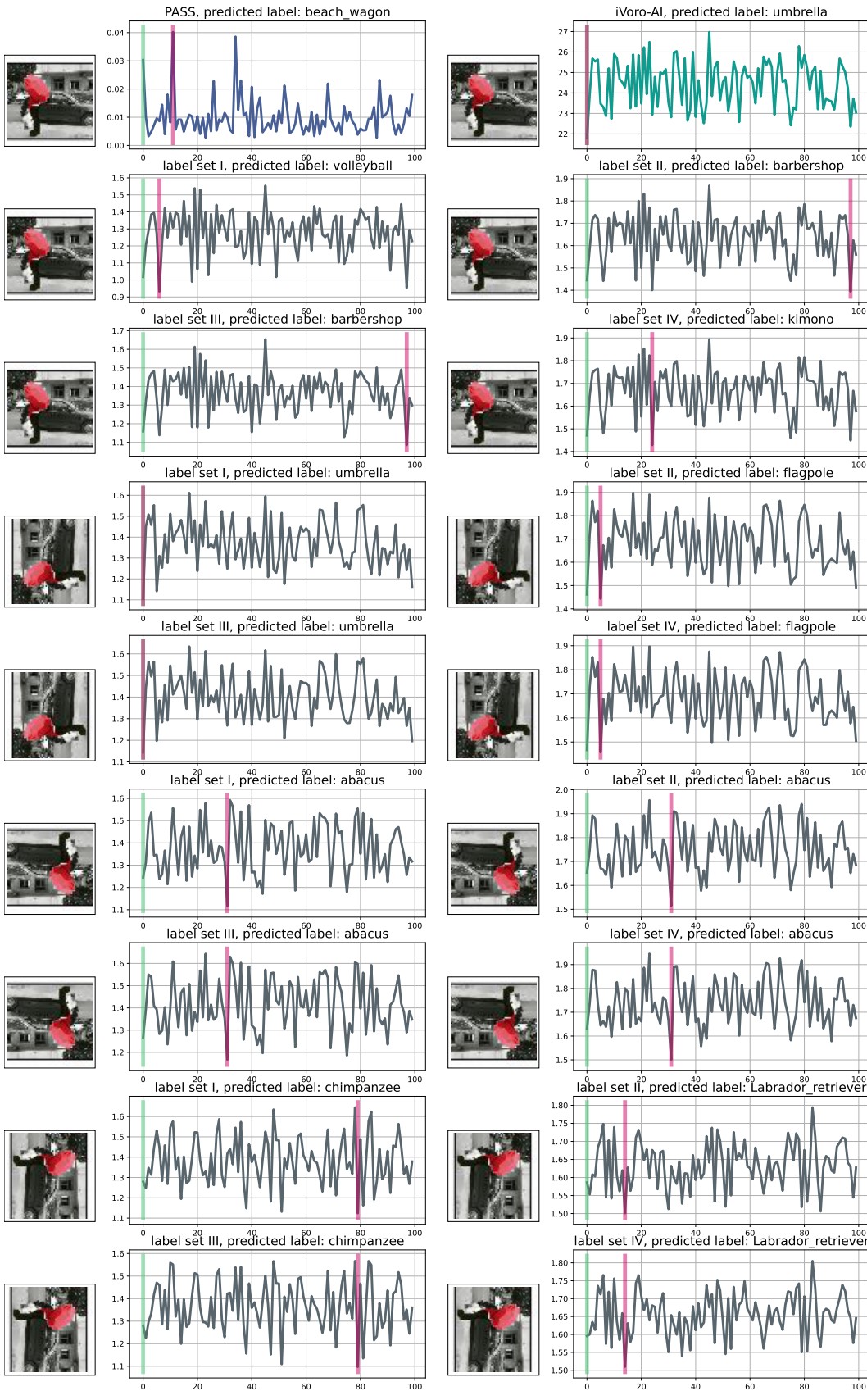

Figure O.2: Sample-level analysis of one example from the "umbrella" class. The y axis denotes the logit for 100 classes for PASS, and denotes the distance to 100 prototypes for iVoro. The green line indicates the ground-truth label while the red line denotes the predicted label.

