# OpenReview forum: "Progressive Voronoi Diagram Subdivision Enables Accurate Data-free Class-Incremental Learning"
_ICLR.cc/2023/Conference — ICLR 2023 poster_

### Official Review · Reviewer_uHig · 2022-10-22

**Confidence:** 3
**Correctness:** 4
**Technical Novelty And Significance:** 3
**Empirical Novelty And Significance:** 3
**Recommendation:** 6

**Clarity, Quality, Novelty And Reproducibility:**

Clarity:
The introduction is rather verbose and could be shortened and improved. It does a good job motivating the need of class incremental learning however it misses some important information, e.g.:

•	In general, the structure of the introduction is a bit confusing. Prior to getting to the contributions part, there is little information about the introduced iVoro models – adding a diagram/Figure of an over iVoro system would make it easier to understand. The details about the models are presented as a part of the contributions’ enumeration – which makes the contribution summary rather lengthy. In general, it might be worth to consider presenting one iVoro system in the introduction as the introduced approach and outline other three in the experimental section as baselines or ablations.

•	Overall, the reviewer finds the contributions rather unclear – consider rewriting the enumerate part of the intro.

•	It might be worth to add a note about the anticipated impact of the introduced model. The authors highlight the strong performance that is nice but adding a few comments on what these results/models change for the research community would strengthen the paper.

•	The authors study “… the challenging data-free CIL problem under the strictest memory and privacy constraints – no stored exemplars and fixed model capacity.” – form the current text it is unclear to the reviewer why this aspect of CIL is worth studying. Adding some motivation would improve the manuscript.

•	The captions of the figures should be self-explanatory and contain all the details required to understand the figures.

•	In Fig. 2 all the scenarios start with different placing of points in the plane – see the left most figure in all scenarios. It would be easier to do across methods comparison if all would start with the same point placings.

•	Methodology section is also hard to follow, it is math heavy and contrails little intuitions on why particular design decisions have been made. Rewriting the methodology part with a focus on presentation simplicity and reproducibility would strengthen the paper.

Quality:
Putting aside the presentation clarity part, the quality of the paper looks good overall. The paper has fair number of experiments and ablations. The reported results show large gains over prior art. Adding standard deviations to reported values would strengthen the quality of the presentation.

Novelty:
The reviewer has not seen before the use of Voronoi Diagram in the context of DNN for class incremental learning, thus there seem to be some edge of novelty. Note, that the reviewer is not actively working on class incremental learning thus might be unaware of some prior art.

Reproducibility:
The authors motion they will release the code on github. However, based on current presentation of the methodology section it might be hard to code up the methods and reproduce the reported results. The reviewer would encourage the authors to rework the methodology section focusing on reproducibility ease.



**Strength And Weaknesses:**

Strengths:
Nice idea of using Voronoi Diagrams in class incremental learning
Rather extensive validation

Weaknesses (details see section below):
Clarity of presentation could be improved
Might be hard to reproduce the models from the current writing of the methodology section


**Summary Of The Paper:**

The paper studies class incremental learning and proposes several approaches based on Voronoi Diagrams called iVoro. In particular, the paper contributes four approaches: iVoro, iVoro-D, iVoro-AC/AI, and iVoro-L. The introduced approaches are validated on three datasets (CIFAR-100, Tiny ImageNet and ImageNet-Subset) reporting large improvements over previous non exemplar class incremental methods.

**Summary Of The Review:**

The introduced ideas are interesting and show good performance in the task of class incremental learning. The paper is well validated and contains significant number of ablations. Overall, the reviewer would lean towards acceptance, assuming that the authors could improve the presentation clarity with a focus on the introduction and methodology sections.

---

> ### Author Response · Authors · 2022-11-18
> **We have systematically rewritten the Introduction and Methodology to enhance readability and reproducibility (with Jupyter notebooks and illustrations).**
>
> We sincerely appreciate your time devoted to reviewing our manuscript! We truly benefit a lot from your invaluable suggestions! We have systematically rewritten the introduction and methodology following all your suggestions, detailed below.
>
> **Q1**: *Clarity and presentation of introduction and methodology.* \
> **A1**: We are sorry if our presentation caused unnecessary difficulty for the readers! In the rebuttal revision, we systematically revised and reorganized the whole manuscript as follows.
> * *"shorten the introduction"*: We have shortened the introduction to improve its conciseness, and added a new section "Our Main Ideas and Results" to help the readers better understand our approach before jumping to the mathematical details.
> * *"add figure about the overall iVoro model"*: Please see Fig. 1 for the overall idea of iVoro, and Fig. 3, Fig. 4, Fig. G.3, and Fig. G.4 for the illustrations of each component.
> * *"contribution summary is lengthy"*: We have rewritten the contribution summary.
> * *"Methodology section is also hard to follow"*: We have added new figures (Fig. G.3, and Fig. G.4) and marked the math notations on the figures to help the readers better understand the mathematical details.
> * *"the anticipated impact of the introduced model"*: We have added "Broader impact" on page 3. We believe that our method can be greatly beneficial not only to the exemplar-free CIL but also to the exemplar-based CIL, which we would leave for future work.
> * *"The captions of the figures should be self-explanatory"*: We have rewritten the captions.
>
> **Q2**: *"...why this aspect of CIL is worth studying."* \
> **A2**: The exemplar-free CIL with fixed model capacity is worth studying because it is the most realistic and practical setting in the real world. \
> **Privacy constraint**: abiding by the privacy policy, the medical (clinical) data should not be shared or distributed without the consent of the patient. Moreover, exemplary medical images are not always available along with the trained model when the model is transmitted across institutions. \
> **Memory constraint**: some recent works consider super long phases (e.g. a thousand phases) for continual learning, in which the memory consumption will be extremely high if the model size keeps increasing.
>
> **Q3**: *"In Fig. 2 all the scenarios start with different placing of points in the plane..."* \
> **A3**: Thanks for your very careful observation! You are definitely right that the positions should be the same in the initial phase, and they actually are the same in (B), (C), and (D). The reason is that (A) fine-tuning did not consider label augmentation, so the distribution of samples is different from the other three. The initial distributions for (B) PASS and (C) iVoro are the same (we clipped the figures to make them consistent with the subsequent phases). It might be somewhat tricky to understand (D) but the distribution is still the same, because in (D) we showed all the expanded classes. If you distinguish the original classes from the expanded ones, they would be identical to (C).
>
> **Q4**: *"Adding standard deviations to reported values..."* \
> **A4**: Yes, you are right that adding standard deviations would strengthen the paper. Please find the robustness analysis on page 22. We run PASS 5 times on CIFAR-100 (10 phases), with mean and standard deviation (std) being 51.80%±2.65%, showing that the previous method may not be very stable. In contrast, iVoro always gives rise to 74.39 because it is deterministic (we use a fixed pretrained model from the first phase).
>
> **Q5**: *Reproducibility.* \
> **A5**: We have uploaded the Jupyter Notebooks containing all the iVoro variants in the Supplementary Material (867-page), to facilitate the reproducibility of our work for the readers. Due to the limited space provided by OpenReview, the complete code, as well as all the pretrained models, feature files, datasets, shell scripts, and instructions will be made publicly available after the reviewing process.
>
> Thank you again for your detailed and constructive suggestions! We hope the above could resolve your concerns, and we would be happy to answer any further questions!

---

> > ### Comment · Reviewer_uHig · 2022-11-22
> > **Thanks for the clarifications**
> >
> > I'd like to thank the authors for the clarifications. After reading the rebuttal and the other two reviews I decided to keep my original rating and recommendation. This is a well executed paper with good improvements over sota methods, with the rebuttal the presentation has improved; however the clarity of the paper could be further polished and improved (e.g. It would be nice to integrate Figs G3 and G4 better within the manuscript -- by moving them to the main body and discussing a bit further.).

---

> > > ### Author Response · Authors · 2022-11-24
> > > **Thanks for your feedback!**
> > >
> > > We are very glad to hear back from you! We will definitely fix this in the next version following your suggestion!

---

### Official Review · Reviewer_K7j8 · 2022-10-25

**Confidence:** 3
**Correctness:** 3
**Technical Novelty And Significance:** 4
**Empirical Novelty And Significance:** Not applicable
**Recommendation:** 8

**Clarity, Quality, Novelty And Reproducibility:**

The text is clear and this work can be reproduced if one has sufficient knowledge in basic machine learning.

**Strength And Weaknesses:**

Strength:
- The paper is clearly written and easy to understand.
- The paper deals with class incremental learning in a new perspective.
- The paper investigates fundamental problem in machine learning.

Weakness:
- I am not quite sure how the method will perform when the distribution of class labels is (highly) skewed.
- The proposed method and theory will work when separation between classes in the feature space is sufficiently clear. The authors mention this as a challenge in the introduction, but I am not quite sure how they alleviated this problem.

**Summary Of The Paper:**

This paper constructs a Voronoi Diagram in an incremental manner to perform Data-free Class-Incremental Learning (CIL). The proposed method is shown to be a flexible, scalable and robust with theoretical insights and experiments, and it promotes the performance of CIL.

**Summary Of The Review:**

I am not an expert in this field so I may be wrong, but based on the papers of my expertise that I have read before, I think this is a good paper in general.

---

> ### Author Response · Authors · 2022-11-18
> **Response to Reviewer K7j8.**
>
> Thank you for your time and effort in reviewing our manuscript! Here we would like to make some clarifications regarding your comments as follows.
>
> **Q1**: *"...how the method will perform when the distribution of class labels is (highly) skewed."* \
> **A1**: Thank you for your question about the skewed distribution. Our experiments are conducted on three standard benchmarking datasets (CIFAR-100, TinyImageNet, and ImageNet-Subset), in which the classes are balanced without out-of-distribution classes. Even though skewed distribution is considered, we believe that our method will still work because our approach imposes no assumption on the data distribution. Due to the page limit of ICLR, we leave imbalanced CIL or CIL with other distributions (e.g. long-tailed distribution) for future work.
>
> **Q2**: *"The proposed method and theory will work when separation between classes in the feature space is sufficiently clear. The authors mention this as a challenge in the introduction, but I am not quite sure how they alleviated this problem."* \
> **A2**: In previous methods, the model keeps updating continuously along the phases, making the old and new features mixed up in the feature space (see Fig. 2A, B). However, in our approach, this issue is significantly alleviated because the model is fixed after the first phase, making all the features from the initial classes fixed as well.
>
> **Q3**: *"...this work can be reproduced if one has sufficient knowledge..."* \
> **A3**: We have uploaded the Jupyter Notebooks containing all the iVoro variants in the Supplementary Material (867-page), to facilitate the reproducibility of our work. Due to the limited space provided by OpenReview, the complete code, as well as all the pretrained models, feature files, datasets, shell scripts, and instructions will be made publicly available after the reviewing process.
>
> We hope this could address your concerns, and please let us know if you have any further questions.

---

> > ### Comment · Reviewer_K7j8 · 2022-11-22
> > **Thanks for the comments**
> >
> > I have gone through the rebuttal and I will stick to my original score.

---

### Official Review · Reviewer_zQrg · 2022-10-31

**Confidence:** 3
**Correctness:** 3
**Technical Novelty And Significance:** 3
**Empirical Novelty And Significance:** 3
**Recommendation:** 6

**Clarity, Quality, Novelty And Reproducibility:**

**Clarity:**
The paper is structured well and the methods are introduced in a clear way. However, the paper suffers in my opinion from the nomenclature of the method itself.

**Quality:**
Literature is quoted in a generous way and experiments are extensive. There is additional experiments and results in the appendix.

**Reproducibility:**
The code will be made available, so the results should be reproducible.



**Strength And Weaknesses:**

**Positive:**
- the paper is structured well and the ideas are presented in a intuitive way
- Illustrations support the main text well
- Extensive citation of the literature
- Code will be made available
- The method shows improvement compared to benchmark methods for CIL

**Negative:**

- (Major) The experimental results show strong two-digit improvement compared to the best baseline (cf. Tab. 1). I missed a section where the authors provide an intuition and reasoning why the method works so much better than everything that has been done before. There is neither a discussion nor a conclusion in “4 Discussion and Conclusion”, more a summary.

- (Minor) It is easy to loose track of the different versions and reading flow suffers from looking up in the text what the different iVoro-x models actually are.

**Questions:**
- It seems the baseline method already shows major improvements. You write *“We suspect that this is because the features generated by the frozen feature extractor can be satisfactorily separable by linear bisectors (Fig. 2). As we can see, the features for other methods are all dramatically changing during the phases, but those for iVoro are all fixed, making incremental VD construction possible. “*  Can you provide a line of argumentation why this could be beyond the above sentence?
- 2.1 - C_t once is the dataset at time step t and once is the set of classes at phase t. If those are different letters I could not distinguish them. The sentence, that C_i are pairwise disjoint irritates because this is not clear.

**Summary Of The Paper:**

This paper presents a method for class-incremental learning, i.e. learning a classification task while during training the model has no access to already used data and new samples can contain new classes. Dividing the classification network in feature extractor and classification head, the authors start with the substitution of the classification head by a 1-nearest-neighbour decision landscape (Voronoi diagram) and derive refined methods from there.
The model is tested on the three established datasets CIFAR-100, TinyImageNet and ImageNet-Subset.
Specifically, the authors develop four components that enhance their baseline model: parametrised normalisation, a divide-and-conquer strategy for iterative construction of decision landscapes, augmentation consensus and integration and multilayer compatibility.
The authors test different combinations of these components to test their respective effects as well as conducting comparison to several benchmark models. The method outperforms all benchmark models in terms of common performance measures on the three datasets by far.

**Summary Of The Review:**

The work at hand is well structured and written. The presented methods are to the best of my knowledge novel in the context of CIL. While the experiments and results stress superiority to other methods, the paper does not discuss its findings sufficiently. An attempt to explain why the method outperforms benchmark models to such a great extent would strengthen the work.

---

> ### Author Response · Authors · 2022-11-18
> **Why does our method achieve such high accuracy? We have added more thorough class-level and sample-level analysis!**
>
> We wholeheartedly appreciate your insightful questions, especially about why our method performs so much better than others. This makes us dive deeper into the experiments and reveal the reason for our surprisingly high performance, as detailed below.
>
> **Q1**: *"...an intuition and reasoning why the method works so much better...", "An attempt to explain why the method outperforms benchmark models to such a great extent would strengthen the work."* \
> **A1**: Thank you for pointing out this critical problem, and this is indeed the major shortcoming in the original manuscript. To intuitively understand the strong two-digit improvement, in the rebuttal revision, we carefully and thoroughly inspect the class-level and sample-level accuracy before and after our method is applied. \
> **Class-level analysis** (pages 40-47): In Appendix N, we illustrate the class-level accuracy along the phases for all the 200 classes in the TinyImageNet dataset. This analysis shows that iVoro-NAC/NAI promotes performance by improving robustness, especially for longer phases. Most of the time, iVoro-NAC/NAI’s performance decreases significantly less than iVoro/PASS during the final several phases. \
> **Sample-level analysis** (pages 48-50): In Appendix O, we arbitrarily inspect two examples from the first class (i.e. "umbrella"), and this reveals the mystery of why iVoro-AI is able to rectify the misclassifications. In the total 16 predictions (4 kinds of rotated images x 4 sets of expanded labels), the mispredicted labels are various (e.g. “ice lolly”, “abacus”, etc.), but the ground-truth label ("umbrella") is always within the top predictions. When integrated together, interestingly, the true label achieves the nearest distance (see upper right figure). This analysis successfully explains how augmentation integration intelligently leverages the SSL-based augmented label sets.
>
> **Q2**: *"...what the different iVoro-x models actually are.", "...the nomenclature of the method..."* \
> **A2**: We are sorry if our nomenclature causes unnecessary difficulty in reading. In the rebuttal revision, we have highlighted the naming convention in the contribution summary (page 3). Generally, the naming is straightforward – N stands for normalization, D for divide-and-conquer, AC/AI for augmentation consensus/integration, and L for multi-layer.
>
> **Q3**: *"...the baseline method already shows major improvements.", "Can you provide a line of argumentation why this could be...".* \
> **A3**: The baseline method (i.e. Voronoi Diagram, 1-nearest neighbor model) has shown competitive or better performance, and this is because the model trained in the first phase is strong enough to provide informative features. To verify this, we gradually decrease the number of the initial classes from 50 to 40, 30, 20, and 10 (see pages 31-32), and the baseline method iVoro performs nearly the same as PASS when the number of the initial classes is too limited (e.g. 10 or 20). Please see Appendix J.
>
> **Q4**: *"...C_t once is the dataset at time step t and once is the set of classes at phase t."* \
> **A4**: In Sec. 2.1, *D_t* is the dataset at time step t, and C_t is the set of classes. We have rewritten this sentence to make it clearer. Sorry for the misunderstanding!
>
> **Q5**: *"The code will be made available..."* \
> **A5**: We have uploaded the Jupyter Notebooks containing all the iVoro variants in the Supplementary Material (867 pages), to facilitate the reproducibility of our work. Due to the limited space provided by OpenReview, the complete code, as well as all the pretrained models, feature files, datasets, shell scripts, and instructions will be made publicly available after the reviewing process.
>
> We hope these additional finer-grained analyses could resolve your concerns. Thank you for your important questions which lead to some interesting new analyses and new findings!

---

> > ### Comment · Reviewer_zQrg · 2022-11-22
> > **Response to Authors**
> >
> > I thank the authors for addressing my comments and keep my initial rating.

---

### Author Response · Authors · 2022-11-18
**Summary of the rebuttal revisions.**

We are very grateful to Reviewers zQrg, K7j8, and uHig for their constructive comments which helped make our paper stronger. We also deeply appreciate that our work is recognized as clearly presented (zQrg and K7j8), novel and interesting (zQrg, K7j8, uHig), with superior performance (zQrg, K7j8, and uHig). In the rebuttal revision, we thoroughly revised the paper considering all reviews. The major changes are indicated by orange text and can be summarized as follows.

1. **Explanation of the superior performance** (zQrg). We have added thorough class-level (pages 40-47) and sample-level (pages 48-50) analysis that clearly interprets how our method improves the performance by collecting and integrating multiple predictions from label augmentation.
2. **Enhancing the presentation of introduction and methodology** (zQrg, uHig). We have shortened the introduction, rewritten the contribution summary, and added illustrations (Fig. G.3, and Fig. G.4) to help the readers understand the mathematical details.
3. **Reproducibility** (zQrg, K7j8, uHig). We have uploaded our Jupyter Notebooks containing all the iVoro variants in the Supplementary Material (867-page), to facilitate the reproducibility of our work.

Thank you all for your comments and suggestions which make our manuscript better! Please kindly let us know if you have any further concerns.

---

### Decision · Program_Chairs · 2023-01-20

**Decision:**

Accept: poster

**Justification For Why Not Higher Score:**

While all reviewers are positive and recommend acceptance, they are not strongly confident, and most of them give weak accepts -- I do not feel comfortable giving the paper a higher score.

**Justification For Why Not Lower Score:**

All reviewers recommend accept and I have no reason to go against their recommendation.

**Metareview: Summary, Strengths And Weaknesses:**

Summary:
This paper presents an algorithm for class-incremental learning utilizing Voronoi diagrams.

Strengths:
- The paper is well written with nice illustrations and a great state of the art section
- Novel perspective on class incremental learning
- Convincing experiments

Weaknesses:
- A discussion section with strengths and weaknesses would make the paper even stronger

**Note From Pc:**

if the above contains the word "oral" or "spotlight" please see: "oral" presentation means -> notable-top-5% and "spotlight" means -> notable-top-25%. As stated in our emails, we are disassociating presentation type from AC recommendations